



# An approach to refining the ground meteorological observation stations for improving PM$_{2.5}$ forecasts in Beijing-Tianjin-Hebei region

Lichao Yang[1], Wansuo Duan[1,2], Zifa Wang[3]

[1]LASG, Institute of Atmospheric Physics, Chinese Academy of Sciences, Beijing, 100029, China

[2]Collaborative Innovation Center on Forecast and Evaluation of Meteorological Disasters (CIC-FEMD), Nanjing University of Information Science and Technology, Nanjing 210044, China

[3]LAPC, Institute of Atmospheric Physics, Chinese Academy of Sciences, Beijing, 100029, China

*Corresponding to*: Wansuo Duan (duanws@lasg.iap.ac.cn)

**Abstract.** The paper investigates how to refine the ground meteorological observation network for greatly improving the PM$_{2.5}$ concentration forecasts by identifying the sensitive areas for targeted observations associated with a total of 48 forecasts in eight heavy haze events during the years of 2016-2018 over the Beijing-Tianjin-Hebei (BTH) region. The conditional non-linear optimal perturbation (CNOP) method is adopted to determine the sensitive area of the surface meteorological fields for each forecast and a total of 48 CNOP-type errors are obtained including wind, temperature, and water vapor mixing ratio components. It is found that, although all the sensitive areas tend to locate within and/or surrounding the BTH region, their specific distributions are dependent on the events and the start times of the forecasts. Based on these sensitive areas, the current ground meteorological stations within and surrounding the BTH region are refined to form a cost-effective observation network, which makes the relevant PM$_{2.5}$ forecasts starting from different initial times for varying events assimilate fewer observations but overall achieve the forecasting skill comparable to, even higher than that obtained by assimilating all ground station observations. This network sheds light on that some of the current ground stations within and surrounding the BTH region are very useless for improving the PM$_{2.5}$ forecasts in the BTH region and can be greatly scattered to avoid the thankless work.



## 1. Introduction

Air pollution has become a serious environmental issue in many Asian countries in recent decades. The

Beijing-Tianjin-Hebei region (BTH region), being one of the most prosperous and populated regions in China, has suffered successive heavy haze events during the past several decades (Xiao et al., 2020). Despite large reductions in primary pollutant emissions due to the recent strict pollution control policies in China, heavy hazy events still occurred in recent years, even during the COVID-19 lockdown period (Huang et al., 2021). The particulate matter of aerodynamic diameter smaller than 2.5 ($PM_{2.5}$) has been

dominated as one of the main air pollutants during the hazy events. Exposure of large population to high $PM_{2.5}$ will pose a higher health risk and even a higher death rate (GBD, 2017; WHO, 2021). Therefore, an accurate prediction of $PM_{2.5}$ concentration is critical for providing early warnings to residents and helping governments take timely actions.

To accurately predict the $PM_{2.5}$ concentrations, it is crucial to improve the quality of meteorological

conditions and emissions since the chemical transport model (CTM) require their information as input. Although the initial chemical concentrations and emission play important roles in air pollution forecasts, the meteorological conditions still substantially influence the $PM_{2.5}$ variations at the regional scale (Liu et al., 2017). In terms of the effect of meteorological initial conditions, lots of studies have shown that small uncertainties in meteorological initial fields will result in large uncertainties in $PM_{2.5}$ forecasts

(Gilliam et al., 2015; Bei et al., 2017). Recently, it has been recognized that a bad meteorological initial condition may even affect the forecast of the accumulation or dissipation processes of the $PM_{2.5}$ event and could result in a false alarm of the heavy haze event (Yang et al., 2022). Therefore, an accurate meteorological initial condition is also crucial for the regional $PM_{2.5}$ forecasts, besides the initial chemical concentrations and emission.

Data assimilation has been recognized as one of the most effective ways to improve the accuracy of initial conditions (Talagrand, 1997). High-quality meteorological initial fields could be obtained by assimilating the observations from an observation network for atmospheric conditions (Snyder, 1996). Among the various meteorological observation sources, the observations from the ground meteorological stations are often assimilated to predict the meteorology fields (Hu et al., 2019; Devers et al., 2020; Yao

et al., 2021). Yang et al. (2022) studied the uncertainties of meteorological initial fields to $PM_{2.5}$ forecasts and found that the meteorological forecasts in the BTH region are much highly sensitive to the

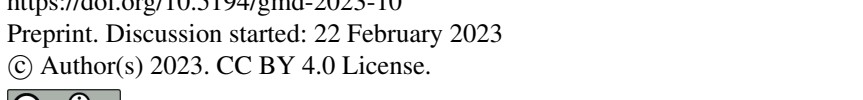
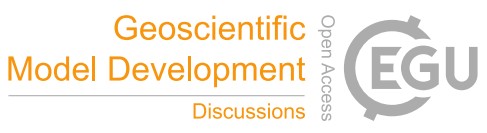

meteorological initial errors at the ground level with the lead time of 12 hours. They emphasized that the initial conditions located at the ground level may play an important role on the meteorological forecasts over the BTH, which will further affect the regional $PM_{2.5}$ forecasts, especially for the forecasts with lead

time of 12 hours. In this sense, assimilating the observations from the ground meteorological stations could make an important contribution to the improvement of the $PM_{2.5}$ forecast skills, especially in the BTH region.

In the past years, high quantity of meteorological stations over the world are constructed to study the atmospheric motions and its weather and climate variabilities; In China alone, there are more than

2000 stations operated by the China Meteorological Administration (CMA) in the year of 2020, whose locations are generally selected based on the administrative district and resident populations (http://data.cma.cn/). Even though there exist a huge number of meteorological stations to provide observations, assimilating more observations may not necessarily lead to much higher forecast benefits. Liu and Rabier (2002) used a simple 1-D framework and the computation of analysis-error covariance to

show that increasing the observation density beyond a certain threshold value would yield little or no improvement in the analysis accuracy. In fact, previous studies have applied both simple and complicated numerical models to argue that additional observations may not result in a large improvement of the forecast skills (Bengtsson and Gustavsson, 1972; Morss et al., 2001; Yang et al., 2014). Theoretically, in the area of strong sensitivity to initial values of the forecasts, assimilating few observations may result

in high forecast skills; conversely, slightly improvements or even worse forecast skills could be resulted even though a large number of observations are assimilated but in the area of weak sensitivity (Janjić et al., 2018; Zhang et al., 2019). Thus, even if we have sufficient meteorological observations, observations in which area and how many observations should be preferentially assimilated to get higher forecasting skills is still a key question. For the ground meteorological stations of concerned here, it is therefore

essential to identify which ones provide the additional observations that dominantly enhance the improvement of the $PM_{2.5}$ forecast level. One of the development targets proposed by the China Meteorological Administration during the 14th Five-Year Plan Period is to arrange the meteorological observation network more reasonably and scientifically (https://www.cma.gov.cn/zfxxgk/gknr/ghjh/202112/t20211208_4295610.html). The results would

provide guidance to refine the existing ground meteorological observation networks for improving the



PM$_{2.5}$ forecasts in the BTH region and avoid the thankless works.

The dominant meteorological stations to be identified, as mentioned above, would provide the meteorological observations that will have the largest impact on the PM$_{2.5}$ forecasts of the concerned region. This idea belongs to the new observational strategy of "targeted observation", that is assimilating

additional observations at the target time $t_1$ in some key areas (i.e. sensitive areas), compared to doing it in other areas, may reduce the forecast errors in the concerned area (verification area) at the future time $t_2$ (verification time; $t_1 < t_2$) to a larger degree. It is obvious that the meteorological stations located in the sensitive areas would provide the meteorological observations that dominantly promote the PM$_{2.5}$ forecasts of the concerned area (i.e. the verification area). Some approaches, such as the singular vector

(SV, Palmer et al., 1998), adjoint sensitivities (Langland et al., 1999), and the ensemble transform Kalman filter (ETKF) (Bishop et al., 2001; Majumdar et al., 2002), have been used to identify the sensitive areas for targeted observations. However, these approaches are developed under the assumption that the initial errors are linearly developed in the nonlinear model, which is not completely true in the real atmosphere (Toth and Kalnay, 1993; Mu and Wang, 2001). In this study, an advanced fully nonlinear method,

Conditional Nonlinear Optimal Perturbation (CNOP; Mu et al., 2003), is applied to seek the initial perturbation of the fastest growth in the nonlinear model and then to determine the meteorological sensitive area of the PM$_{2.5}$ forecasts. It has been verified that the sensitive area identified by the CNOP shows advantages compared with the areas identified by traditional methods through both the theoretical proves and numerical experiments (Qin and Mu, 2011; Chen et al., 2013; Duan et al., 2018). The CNOP

has been adopted to identify the sensitive areas in the studies of tropical cyclones, El Nino-Southern Oscillation events, oceanic mesoscale eddies, and marine environments and has successfully improve the forecasting skills (see the review of Duan et al., 2022). Especially, Yang et al. (2022) applied the CNOP to determine the sensitive areas for targeted observation of a heavy hazy event which was not warned in time by the monitoring center and demonstrated that assimilating additional observations in such

sensitive area leads to successful forecast of the PM$_{2.5}$ concentrations with much higher skill. Then in this study, we would use the CNOP to recognize the dominant ground meteorological stations applicable for PM$_{2.5}$ forecasts by investigating the sensitive areas of eight winter heavy hazy events over the BTH region during years of 2016-2018, consequently providing an idea to refine the current ground meteorological stations for improving the PM$_{2.5}$ forecasts in the BTH. It is noted that during this period,





encouraged by the strict pollution control policies issued by the Chinese government, great efforts have been made to produce more accurate high-resolution emission inventory (Zheng et al., 2020), which is favorable for better simulating the chemical components in China and then separating the meteorological uncertainty effects of interest in the present study.

The remainder of the paper is organized as follows. In Sect.2, we introduce the model, data and
method. In Sect.3, we reproduce the eight heavy hazy events occurred in the BTH during 2016-2018 and identify the sensitive areas of surface meteorological conditions for the PM$_{2.5}$ forecasts with the application of CNOP method. Then a cost-effective meteorological observation network is constructed in Sect. 4, which has been verified to be an approximation to the whole BTH ground meteorological stations for improving the PM$_{2.5}$ forecasts. In Sect. 5, we interpret the reasons why assimilating the cost-
effective observations can lead to an improvement of the PM$_{2.5}$ forecast skill comparable to assimilating the whole ground observations from the perspectives of thermodynamics and dynamics, and in Sect.6, a summary and discussion is finally provided.

## 2. Model, Data and Method

In this study, we use the Weather Research and Forecasting model (WRF) and its adjoint model, and the
Nested Air Quality Prediction Modeling System (NAQPMS) to identify the sensitive areas of surface meteorological conditions associated with the regional PM2.5 forecasts by the application of CNOP approach.

### 2.1 Models

The NAQPMS model is a 3-D regional Eulerian chemical transport model, which contains emissions,
advection/convection, diffusion, dry and wet deposition, gas/aqueous chemical modules (Wang et al., 1997; 2006). It has been widely used in scientific studies and practical forecasts for air quality in China. The anthropogenic emissions are obtained from Multi-resolution Emission Inventory for China (http://meicmodel.org/). Since in the present paper, we only focus on the sensitivity of meteorological conditions on PM$_{2.5}$ forecast, the emission inventory is assumed as perfect and is kept as the same in all
the simulations. The modelling domain includes 119×119 grids with a horizontal resolution of 30km and 20 levels in the vertical.



The NAQPMS model is driven by the meteorological fields generated through the WRF (http://www.wrf-model.org/). The parametrization schemes adopted in the WRF model include the Lin microphysics scheme (Lin et al. 1983), Dudhia shortwave radiation schemes (Dudhia, 1989), RRTMG

longwave radiation (Iacono et al. 2008), and Yonsei University planetary boundary layer parameterization scheme (Hong et al. 2006). The adjoint model of WRF also use the same parameterization schemes.

## 2.2 Data

There are eight typical heavy hazy events occurring in the BTH region during the wintertime (OND,

October-November-December) in the years of 2016-2018 (Table 1) and all these eight events and their associated forecasts are concerned in the study. The observed surface $PM_{2.5}$ concentration datasets of the events are obtained by the national environmental monitoring stations. Exactly, there are 80 air quality monitoring stations within the BTH region [see the geographical distribution for these 80 stations in Fig. 1(a)]; and from these stations, we retrieved the hourly $PM_{2.5}$ concentration time series for each of the

eight events.

To produce the initial and boundary conditions for WRF simulation, the fifth generation ECMWF reanalysis for the global climate and weather (ERA5, https://www.ecmwf.int/en/forecasts/datasets/ reanalysis-datasets/era5 ) and National Centers for Environmental Prediction (NCEP) GFS historical archive forecast data (GFS, https://rda.ucar.edu/ datasets/ds084.1/) are used.

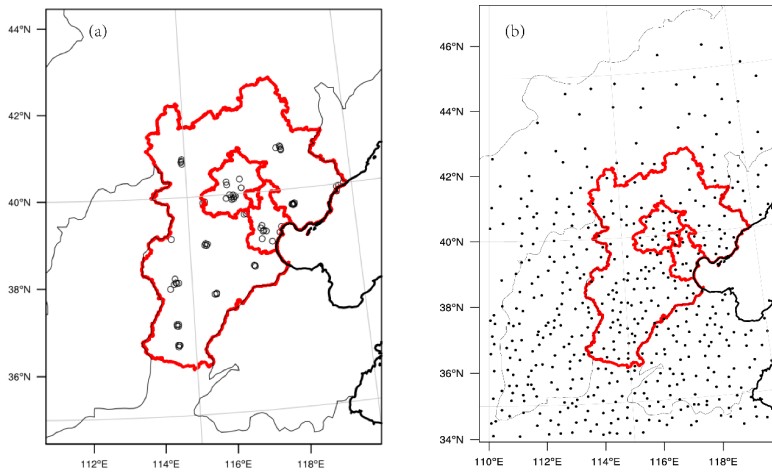


**Figure 1 The maps of (a) the 80 environmental monitoring stations (black circles) within the BTH region and (b) the 481 national ground meteorological stations (black dots) within and surrounding the BTH region**



**([110ºE~120ºE, 34ºN~46ºN]). The black lines represent the boundaries of provinces in China, and the thick black lines are the coastline. The boundaries of the Beijing City, Tianjin City and Hebei Province are presented in thick red lines.**

### 2.3 Conditional Nonlinear Optimal Perturbation (CNOP)

The CNOP represents the initial perturbation (or initial error) that result in the largest forecast error in the verification area at the verification time and is the most sensitive initial perturbation. The dynamical equation in the nonlinear model can be written as Eq. (1),

$$\begin{cases} \frac{\partial x}{\partial t} + F(\boldsymbol{x}) = 0 \\ \boldsymbol{x}|_{t=0} = \boldsymbol{x}_0 \end{cases}, \qquad (1)$$

where $t$ is the time, $F$ is the nonlinear partial differential operator and $\boldsymbol{x}$ is the state vector with an initial value $\boldsymbol{x}_0$. If we add an initial perturbation $\delta \boldsymbol{x}_0$ to the initial state $\boldsymbol{x}_0$, the evolution of the two initial states at the prediction time $T$ can be described as Eq. (2),

$$\boldsymbol{x}(T) = M(\boldsymbol{x}_0), \boldsymbol{x}(T) + \delta \boldsymbol{x}(T) = M(\boldsymbol{x}_0 + \delta \boldsymbol{x}_0), \qquad (2)$$

where $M$ is the nonlinear propagator that propagates the initial value to the prediction time $T$. So $\delta \boldsymbol{x}(T)$ describes the evolution of initial perturbation $\delta \boldsymbol{x}_0$ of the reference state $\boldsymbol{x}(T)$. An initial perturbation is called as CNOP $(\delta \boldsymbol{x}_0^*)$ if and only if

$$J(\delta \boldsymbol{x}_0^*) = \max_{\delta \boldsymbol{x}_0{}^T C_1 \delta \boldsymbol{x}_0 \le \beta} [M(\boldsymbol{x}_0 + \delta \boldsymbol{x}_0) - M(\boldsymbol{x}_0)]^T C_2 [M(\boldsymbol{x}_0 + \delta \boldsymbol{x}_0) - M(\boldsymbol{x}_0)]. \qquad (3)$$

The $\delta \boldsymbol{x}_0^T C_1 \delta \boldsymbol{x}_0 \le \beta$ is the constraint condition of initial perturbation and $\beta$ is a positive value. $C_1$ and $C_2$ are coefficient matrices, which define the format of the initial perturbation and its evolution. Mathematically, the CNOP leads to the global maximum of the cost function $J(\delta \boldsymbol{x}_0^*)$ under the certain constraint.

In our study, since we focused on the uncertainties of meteorological initial condition associated with the PM$_{2.5}$ forecast, following Yang et al., (2022), the state vector $\boldsymbol{x}$ consists of zonal and meridional wind (U and V, respectively), temperature (T), water vapor mixing ratio (Q) and pressure (P) components, which are considered as important meteorological fields on PM$_{2.5}$ forecasts over the BTH region (see the review paper of Chen et al., 2020). The perturbations $\delta \boldsymbol{x}_0$ are superimposed on the ground meteorological field $\boldsymbol{x}_0$ of interest. The amplitude of initial perturbation and its evolution are defined by the total energy of meteorological state at the ground level of the model domain and the integral of the total energy from ground to top (i.e., 100 hPa) at the verification areas (i.e. the BTH region), respectively. The expression of total energy is shown in Eq. (4) (Ehrendorfer et al., 1999),



$$\text{Total energy} = U^2 + V^2 + \frac{C_p}{T_r}T^2 + \frac{L^2}{C_p T_r}Q^2 + R_a T_r (\frac{P}{P_r})^2, \quad (4)$$

where $C_p$ (=1005.7 Jkg$^{-1}$ K$^{-1}$), $R_a$ (=287.04 Jkg$^{-1}$ K$^{-1}$), $T_r$ (=270 K), $L(= 2.5105 \times 10^6$ Jkg$^{-1}$) and $P_r$ (=1000 hPa) are constant values.

The spectral projected gradient 2 (SPG2) method is used to solve the optimization problem in Eq. (3). It is noted that the SPG2 algorithm is generally designed to solve the minimum value of nonlinear function (cost function) with an initial constraint condition, and the gradient of cost function with respect to the initial perturbation represents the descending direction of searching for the minimum of the cost function. Therefore, in this study, we have to rewrite the cost function Eq.(3) as $J'(\delta x_0^*) = \min\limits_{\delta x_0{}^T C_1 \delta x_0 \le \beta} -$

$[M(x_0 + \delta x_0) - M(x_0)]^T C_2 [M(x_0 + \delta x_0) - M(x_0)]$ and the WRF adjoint model is used to compute the gradient of the cost function. For further details of the SPG2 algorithm, please refer to Birgin et al. (2001).

### 3.    The sensitive areas of surface meteorological field for the PM$_{2.5}$ forecasting

In this section, we first simulate the PM$_{2.5}$ concentrations variability using the WRF initialized by the

ERA5 reanalysis data and NCEP-GFS forecast data separately to show the sensitivities of PM$_{2.5}$ forecasts to the meteorological initial uncertainties. Then we calculated the CNOP-type initial errors of concerned forecasts and identify their sensitive areas.

#### 3.1  Sensitivity to meteorological initial uncertainties of PM$_{2.5}$ variability simulations.

For each of the eight heavy haze events, after the 10-day spin-up of WRF-NAQPMS, the ERA5 and the

GFS data are separately used to initialize the WRF model and then two forecasted meteorological fields can be obtained, which force the NAQPMS to output two kinds of simulations of PM$_{2.5}$ concentrations. Table 1 provides the initial and final times of the eight events simulations and Fig. 2 plots the two kinds of simulations of the PM$_{2.5}$ concentrations averaged over the BTH region for each event and corresponding observations. We take the event initialized at 00:00 BJT on 30 November 2018 as an

example to describes the difference between two kinds of simulations of PM$_{2.5}$ concentrations (see Fig. 3). Exactly, for this event, the ERA5 presents weak southerly winds with a mean speed of 1.06 ms$^{-1}$ over the BTH region at the initial time, while the GFS shows stronger southerly winds with the speed of 1.91 ms$^{-1}$. Obviously, the two simulations show a difference in the initial meteorological fields of this event.



When the time comes to the final time after 18 hours, the simulation initialized by ERA5 presents weak northerly wind in the BTH region and forecast the $PM_{2.5}$ concentration of 93.05 $\mu g\ m^{-3}$ averaged over the BTH region; however, the simulation initialized by GFS enhances the southerly wind to 3.56 ms$^{-1}$, and particularly in the southern part of Hebei the southerly wind reaches to 5.89 ms$^{-1}$, which transports more $PM_{2.5}$ from the south to the BTH region and result in the $PM_{2.5}$ forecasts of 134.71 $\mu g\ m^{-3}$ on average. It is noted that these two $PM_{2.5}$ simulations are generated from the same emission inventory and

the same initial chemical concentrations, with the initial $PM_{2.5}$ concentration concentrating in Anhui and Hubei provinces, which are located to the south of BTH region. It is therefore certain that the difference between the two $PM_{2.5}$ simulations of this illustrated event are only caused by the different meteorological initial fields. For other forecasts, it is also seen from Figure 2 that different initial meteorological conditions result in different $PM_{2.5}$ simulation accuracy, in terms of the magnitude, peak time and even

the variability in the accumulation and dissipation processes of the heavy haze event.

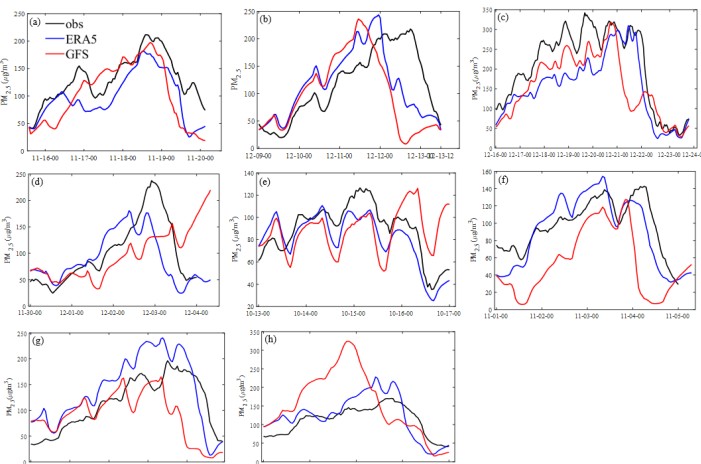

**Figure 2 Time series of the $PM_{2.5}$ concentrations averaged over the BTH region of observations (black line) and simulations initialized by ERA5 (blue line) and GFS (red line) meteorological data during the eight heavy hazy events in 2016-2018. These events occurred during (a) 12:00BJT, 15 Nov-02:00 BJT, 20 Nov in 2016; (b)**

**00:00 BJT, 9 Dec-12:00 BJT, 13 Dec in 2016; (c) 00:00 BJT, 16 Dec-00:00 BJT, 23 Dec in 2016; (d) 00:00 BJT, 30 Nov-00:00 BJT, 4 Dec in 2017; (e) 00:00 BJT, 13 Oct-00:00 BJT, 17 Oct in 2018; (f) 00:00 BJT, 1 Nov-00:00 BJT, 5 Nov in 2018; (g) 00:00 BJT, 11 Nov-00:00 BJT, 16 Nov in 2018; (h) 00:00 BJT, 30 Nov-00:00 BJT, 4 Dec in 2018**.

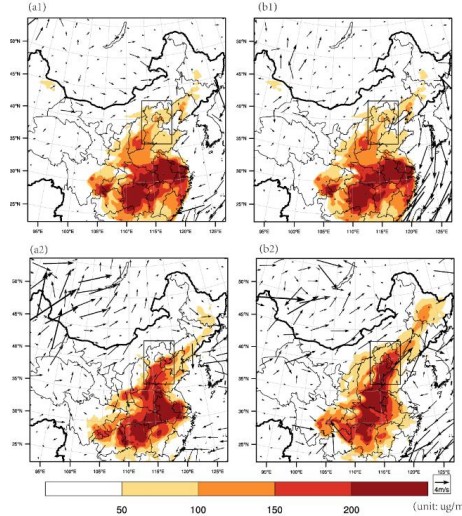

**Figure 3 The surface wind (vector, m s⁻¹) and PM₂.₅ concentration (shaded; μg m⁻³) components of the initial states for the simulation of the event during 30 November and 4 December in 2018 (1) and their evolutions at the lead time 18 hours (2), where the initial time is 00:00 30 November 2018 and (a) is initialized by the ERA5 and (b) is initialized by the GFS.**

To quantify the different sensitivities of the two simulations on initial meteorological conditions,

the Root Mean Square Error (RMSE) and correlation coefficients (CC) between the simulation and observation of the eight events are calculated. It is found that, of all the eight events, the ERA5 simulations show smaller RMSEs and higher CCs with respect to the observations (see Table 1). If we take an average of the eight events for the whole simulation period (see Table 1 and Figure 2), the RMSE of the ERA5 and GFS simulations are 41.16 μg m⁻³and 59.83μg m⁻³, respectively; the CCs of the

ERA5 and GFS simulations reach to 0.79 and 0.50, respectively. Thus, for all the heavy hazy events considered, the simulations initialized by ERA5 reanalysis perform better than the GFS forecast data. In fact, since the ERA5 reanalysis data was obtained by assimilating all available observations with more advanced model by ECWMF, it has much high quality and often regarded as an approximation to the real atmosphere. It is therefore comprehensible that the ERA5 performs much well in simulating the PM₂.₅

concentrations. This also indicates that the PM₂.₅ forecasting uncertainties made by the WRF-NAQPMS are highly sensitive to meteorological initial conditions and a much accurate meteorological initial condition is essential for PM₂.₅ forecasts. Although the simulations initialized by ERA5 reanalysis perform better than the GFS forecast data, they still depart from the observations. Therefore, considering





the sensitivity of meteorological field accuracy on $PM_{2.5}$ concentration simulations, it is necessary to

identify the sensitive area of the meteorological initial field for $PM_{2.5}$ forecasts and assimilate additional

targeted observations, further pushing the $PM_{2.5}$ simulation resulted by the ERA5 much closer to the truth.

**Table 1. The RMSE ($\mu g\ m^{-3}$) and CC of $PM_{2.5}$ concentrations between the simulations initialized by the ERA5 / GFS and the observations in the eight heavy hazy events. The simulation of smaller RMSE and higher CC**

**is marked in bold.**

| Cases | Initial time / Final time (BJT, Day Month, Year) | RMSE (ERA5/GFS) | CC (ERA5/GFS) |
|---|---|---|---|
| 1 | 1200, 15 Nov / 0200, 20 Nov, 2016 | **39.66**/44.21 | **0.86**/0.78 |
| 2 | 0000, 09 Dec / 1200, 13 Dec, 2016 | **56.94**/82.06 | **0.57**/0.25 |
| 3 | 0000, 16 Dec / 0000, 23 Dec, 2016 | **66.89**/72.42 | **0.91**/0.84 |
| 4 | 0000, 30 Nov / 0000, 04 Dec, 2017 | **47.46**/51.91 | **0.64**/0.56 |
| 5 | 0000, 13 Oct / 0000, 17 Oct, 2018 | **14.87**/27.29 | **0.86**/0.16 |
| 6 | 0000, 01 Nov / 0000, 05 Nov, 2018 | **20.74**/53.88 | **0.86**/0.59 |
| 7 | 0000, 11 Nov / 0000, 16 Nov, 2018 | **45.94**/61.42 | **0.83**/0.28 |
| 8 | 0000, 30 Nov / 0000, 04 Dec, 2018 | **36.77**/85.45 | **0.81**/0.54 |

### 3.2 The sensitive areas of meteorological initial fields for $PM_{2.5}$ forecasts

From Figure 2, it is known that when the haze started to develop, it usually takes more than 2 days to

accumulate and dissipate rapidly in a few hours. For example, for the event occurred during the period

from 00:00 BJT (Beijing Time, UTC+8 hours) on 9 Dec to 12:00 BJT on 13 Dec in 2016, the haze started

to accumulate at approximately 20:00 on 9 Dec and it took 55 hours to accumulate $PM_{2.5}$ from 45 $\mu g\ m^{-3}$

to 208 $\mu g\ m^{-3}$; this high $PM_{2.5}$ concentration sustained for almost 16 hours, then from 18:00 on 12 Dec,

the $PM_{2.5}$ concentration decreased from 217 $\mu g\ m^{-3}$ to 46 $\mu g\ m^{-3}$ in 18 hours. Certainly, the stable

atmospheric boundary layer will lead to the accumulation of $PM_{2.5}$ concentrations, while the dissipation

is mostly attributed to the large winds or wet deposition (Chen et al., 2020). These distinct mechanisms

may indicate that the sensitive areas of meteorological initial field are different for the $PM_{2.5}$ forecasts

during the accumulation and dissipation processes. Yang et al. (2022) investigated the vertical energy



profiles of the most sensitive meteorological initial perturbations (i.e. the CNOP-type error) of the PM$_{2.5}$

forecasts in one heavy haze event in the BTH, and they showed that, for the forecasts during either

accumulation or dissipation processes, the large energy of the CNOP-type errors mainly lie at the low

level of the atmosphere for the lead time of 24 hours and at the ground level for the lead time of 12 hours.

It is indicated that the uncertainties of ground meteorological initial conditions may play a more

important role on the PM$_{2.5}$ forecasts with the lead time of 12 hours. To further assess the role of ground

meteorological initial fields on the PM$_{2.5}$ forecasts, we calculated the CNOP-type errors for the eight

heavy haze events in this study, as Yang et al. (2022) did, and found that the PM$_{2.5}$ forecast uncertainties

are indeed much sensitive to the accuracy of ground meteorological initial conditions for the lead time

12 hours [The details are omitted here because of similar thoughts to Yang et al. (2022)]. This result,

relative to the economic property of the target observation strategy (see the introduction), inspires us to

investigate the current ground meteorological stations within and surrounding the BTH and to see if they

can be refined to more cost-effectively improve the PM$_{2.5}$ forecasts in the heavy haze events by exploring

the sensitive areas of ground meteorological fields forecasting.

To do it, we consider the forecasts with the fixed lead time of 12 hours but with different start times.

For each event we analyze 4 cycle forecasts every 12 hour from its start time (see Table 2) over the

accumulation process (hereafter as AFs) and 2 forecasts over the dissipation process (hereafter as DFs).

As a result, a total of 32 AFs and 16 DFs were obtained for the eight events of investigation. To identify

the sensitive areas of the ground meteorological field in each forecast, we adopt the idea of Lorenz (1965)

that when exploring the effect of initial error growth, an assumption of perfect model is done. However,

in reality, whichever it is initial filed of model, even emission inventories, it certainly consists of

uncertainties. So to make much realistic we have to take the better simulation initialized by ERA5 as

"truth run" because we cannot obtain relevant observations from the Monitor center for assimilations and

the worse simulation initialized by GFS forecast data as "control forecast" or "control run". The

differences between them reflect the sensitivities of forecast uncertainties of PM$_{2.5}$ concentrations on the

accuracy of initial meteorological field. Therefore, when one computes the CNOP-type initial

perturbation superimposed on the better simulation initialized by ERA5 (i.e. "truth run"), it can be

regarded as an approximation to the most sensitive initial error that disturbs the meteorology forecast of

the BTH region and then the PM$_{2.5}$ forecast result. According to this perturbation, we can determine the


sensitive area of the meteorological field (see next subsection) and preferentially assimilating additional observations in the sensitive area of the control forecast will make the updated forecast (hereafter as "assimilation run") approach to the truth run (see Yang et al., 2022). Such idea is a kind of observation

system simulation experiment (OSSE, Masutani et al., 2020). It is conceivable that, if the real observations are available, assimilating the real observations on the sensitive areas of ERA5 simulation will also make the ERA5 simulation much closer to the real truth. In our study, we adopt this idea to determine the sensitive areas. Since the real observations are not in public archive, the "additional observations" are correspondingly taken from the initial field of the truth run (i.e. the ERA5 data) and

called as "simulated observations" according to the OSSEs. These simulated observations include the wind, temperature and relative humidity variables and they are all the standard meteorological variables monitored in the national meteorological stations; and the relevant assimilations are performed by the WRF-3DVar schemes.

**Table 2.   Start times of the cycling AFs and DFs for the eight heavy hazy events.**

| Cases | AFs (BJT, Day Month, Year) | DFs (BJT, Day Month, Year) |
|:---:|:---:|:---:|
| 1 | 02:00, 16 Nov, 2016 | 14:00, 18 Nov, 2016 |
| 2 | 14:00, 09 Dec, 2016 | 02:00, 12 Dec, 2016 |
| 3 | 14:00, 16 Dec, 2016 | 02:00, 22 Dec, 2016 |
| 4 | 14:00, 30 Nov, 2017 | 20:00, 02 Dec, 2017 |
| 5 | 14:00, 13 Oct, 2018 | 20:00, 15 Oct, 2018 |
| 6 | 14:00, 01 Nov, 2018 | 02:00, 04 Nov, 2018 |
| 7 | 20:00, 11 Nov, 2018 | 20:00, 14 Nov, 2018 |
| 8 | 02:00, 30, Nov, 2018 | 20:00, 02 Dec, 2018 |


Now we determine the sensitive areas of the ground meteorological field associated with $PM_{2.5}$ forecasts in the BTH. For this purpose, the CNOP-type initial errors superimposed on the ground meteorological fields are calculated for each of the 48 $PM_{2.5}$ forecasts with the application of WRF and its adjoint model by using the SPG2 solver (see section 2). Then a total of 48 CNOP-type initial errors

are obtained for the 48 forecasts including 32 AFs and 16 DFs. For the AFs, the CNOP-type errors



basically concentrate within the BTH region, although there exist position differences among the forecasts; while for the DFs, the CNOP-type errors are mostly located on the northern part of the BTH region, but the specific structures are dependent on the start time. Figure 4 shows two examples of the CNOP-type errors with the wind and temperature components during AF and DF of the heavy haze event

occurring during 1-5th November 2018, respectively. It can be seen that, for the AF started from 02:00 on Nov 2 in 2018, the CNOP-type error presents large southerly wind anomalies at the southern part of the BTH region, particularly in the cities of Anyang and Liaocheng, and large negative temperature anomalies almost located within the BTH region; and for the AF started from 14:00 on Nov 2 in 2018, the large southerly wind errors are dominant in the Jining city of Shandong province, while the negative

temperature error concentrate in the southern part of Hebei region; as for the two examples of the CNOP-type errors of the DFs, one is for the forecast initialized at 02:00 on Nov 4 in 2018 and exhibits large northerly wind and negative temperature anomalies on the northern part of BTH region, covering the region of Abaga Banner, with the much large temperature anomalies in the southern part of Shandong Province; the other is for the forecast at the 02:00 on Nov 5 in 2018 and presents the northerly winds and

negative temperature anomalies over the northern part of Hebei province. It is obvious that the CNOP-type errors, though they are all mainly presented around the BTH region, provide different areas where different meteorological variable errors concentrate even for the same forecast. To overcome this embarrassment, we evaluate the total moist energy norm (TME; Yang et al., 2022) of the CNOP-type errors.

$$\text{TME} = \frac{1}{2}\left(U'^2 + V'^2 + \frac{c_p}{T_r}T'^2 + \frac{L^2}{c_p T_r}Q'^2 + R_a T_r(\frac{P'}{P_r})^2\right). \quad (5)$$

The TME considers all the concerned meteorological variables in the CNOP-type errors and measures the comprehensive sensitivity of $PM_{2.5}$ forecast uncertainties on initial meteorological perturbations. Then the $PM_{2.5}$ forecasts are more sensitive to the combined effect of all meteorological variables' uncertainties occurred in the area with larger values of TME and these areas are regarded as the sensitive

areas (see Yang et al., 2022). Figure 5 shows the spatial distribution of the TME for the 4 forecasts mentioned above. It is seen that, for the two AFs, their sensitive areas (i.e. the areas with larger values of TME) are mostly located in the BTH region, especially in the Beijing City and southern part of Hebei province; but for the forecast started from 02:00 on Nov 2 in them, the area in the center of Shandong province is also additionally denoted as a sensitive area. For the two DFs, their sensitive areas, compared



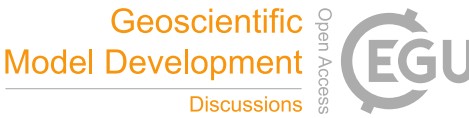

with those of the two AFs, move northward and the one in the forecast initialized at 02:00 on Nov 4 is

mostly located in the Inner-Mongolia and western part of Liaoning provinces, while the other forecast

presents its sensitive area closer to the BTH region, mostly located in the cities of Chengde and

Zhangjiakou in Hebei province.

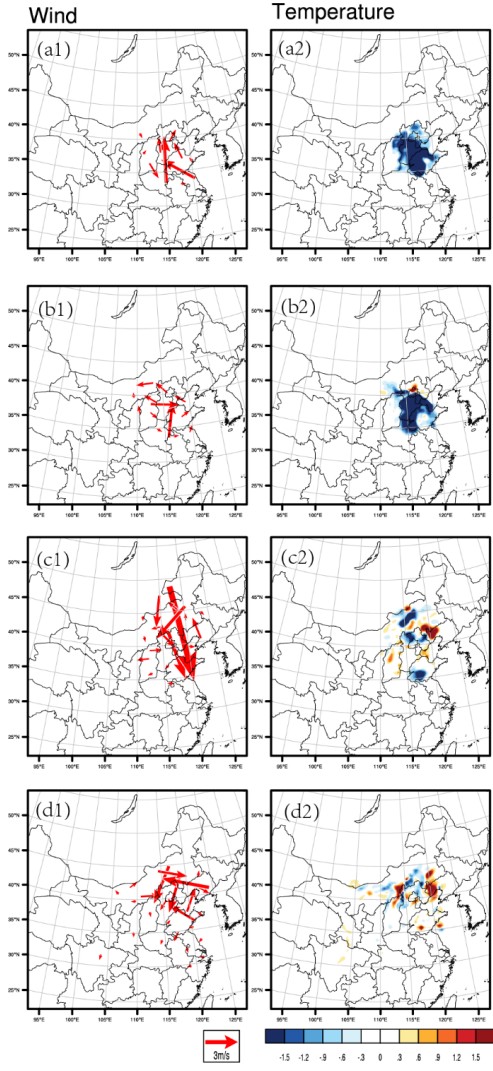


**Figure 4 The horizontal distribution of the wind (1) and temperature (2) components of the CNOP-type errors for the AF started from the 02:00 on Nov 2 (a) and from 14:00 on Nov 2 in 2018 (b), and for the DF started from 02:00 on Nov 4 (c) and from 14:00 on Nov 4 in 2018 (d).**

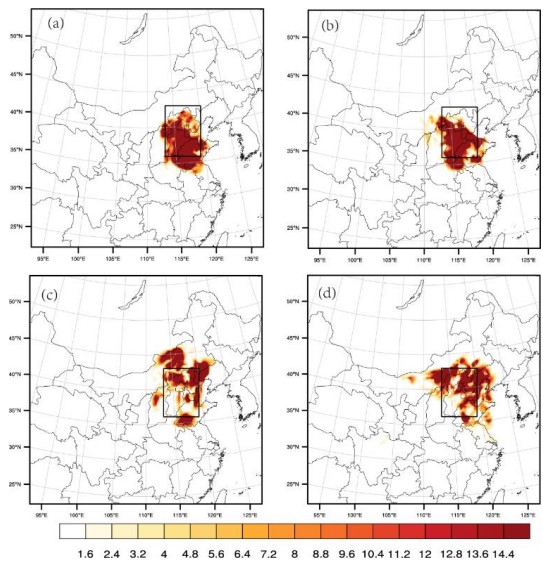


**Figure 5 The horizonal distribution of the TME (unit: J kg⁻¹) for the AFs started from (a) 02:00 on Nov 2 and (b) 14:00 on Nov 2 in 2018, and for the DFs started from (c) 02:00 on Nov 4 and (d) 14:00 on Nov 4 in 2018. The black rectangle is the verification area, i.e. the BTH region.**

From the sensitive areas above, it is easily known that, even for the same event, the specific distributions of the sensitive areas are dependent on the start times of the forecasts. It is therefore conceivable that the 48 forecasts for the eight events will exhibit the sensitive areas of multifarious structures and locations. In terms of this situation, one naturally asks that how to reveal a cost-effective observation network that does for the $PM_{2.5}$ forecasts starting from different initial times for different events. Relative to the ground meteorological stations in China of interest, the above question can be

converted into how to refine the current meteorological stations within and surrounding the BTH and make them applicable for improving much cost-effectively the $PM_{2.5}$ forecasts with different start times for different heavy haze events. This question will be addressed in the next section.

### 4. The cost-effective meteorological observation network applicable for significantly improving the $PM_{2.5}$ forecasts

In this section, we will construct the cost-effective meteorological observation network based on the sensitive areas identified by the CNOP-type errors of the 48 forecasts for the eight heavy haze events; then a series of OSSEs (see section 3.2) are conducted to show the advantage of the additional





observations from this observational network in improving the PM$_{2.5}$ forecasting skills, which finally

provides a strategy to refine the current meteorological stations within and around the BTH.

**4.1 An essential observational network that enhances the PM$_{2.5}$ forecasting skill much greatly**

For the 48 CNOP-types errors, we use a quantitative frequency method [see Duan et al. (2018)] to identify

the spatial grids that are often covered by large values of the TME. Specifically, for each CNOP-type

error, we sort its spatial grid points with a decreasing order according to the amplitude of the TME and

choose the first 3% grid points of the model domain; then a total of 424 grid points is obtained, which

bear larger TME values than other grid points and contribute more to the meteorological forcing errors

associated with the relevant PM$_{2.5}$ forecast (see Yang et al., 2022). Note that we select here the first 3%

grid points so that the number 424 of sensitive grid points is close to the number 481 of the current

meteorological stations within and surrounding the BTH (see Fig. 1b), in attempt to investigate whether

the sensitive grid points explain the current ground stations. Since 32 AFs are considered in the study,

we can get 32 grid point sequences from their 32 CNOP-type errors and in each sequence, there are 424

grid points. For each grid point, we compute its frequency of each grid $(i, j)$ occurring in the 32 sequences

by the Eq. (8).

$$F_{i,j} = \frac{c_{i,j}}{N} \times 100\%, \quad (6)$$

where $c_{i,j}$ is the number of the grid point $(i, j)$ occurring in all sequences and $N$ denotes the number of

all sequences (here is 32). We define a threshold 60% and select the grid points with $F$ larger than 60%,

which means that the grid point $(i, j)$ exists in most of the sequences. Then a total of 174 grid points is

determined. These 174 grid points are certainly frequently carrying much large meteorology errors

measured by the TME in the 32 CNOP-type errors for the 32 AFs. Similarly, we also obtain 184 grid

points from the 16 CNOP-type errors of the 16 DFs (Fig. 6a, b). We incorporated the 174 grid points for

the AFs and the 184 grid pints for the DFs into an integrated observation network, as compared with the

current ground meteorological stations that has been constructed within and surrounding the BTH region

(110E~120E, 34N ~46N, see Fig. 1b). It is found that the meteorological stations have been constructed

with 99 ones in the area covered by the 174 grid points for the AFs and with 60 ones in the area covered

by the184 grid points for the DFs. Since these 99 stations for AFs and 60 stations for DFs, a total of 127

stations (32 stations are overlapped), are all located in the area covered by the sensitive 174 grid points

for AFs and 184 grid pints for DFs, they could provide additional observations that help improve much

significantly the skill of the $PM_{2.5}$ forecast in the BTH, as compared with other constructed stations but

not in the sensitive grids. For this reason, we regard the network spanned by these 127 stations as an

"essential network" [see Fig. 6(c)].

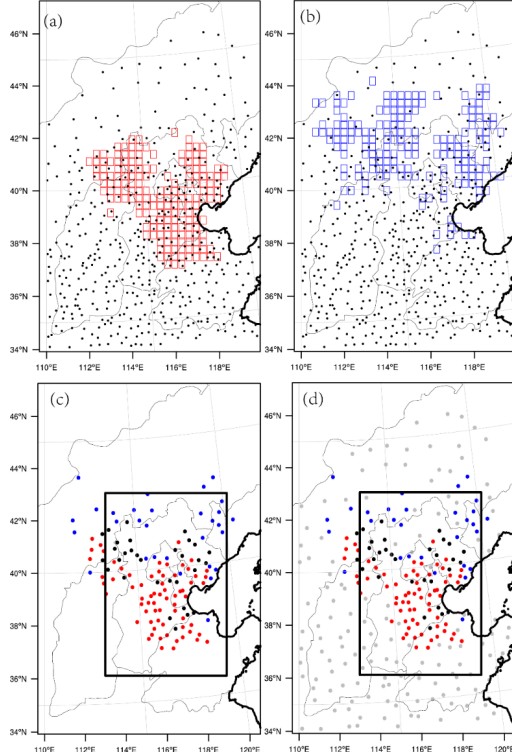


**Figure 6 The spatial distributions of the 174 sensitive grid points (red squares) for AFs (a), the 184 sensitive grid points (blue squares) for DFs (b), and all constructed stations [denoted by black dots in (a) and (b)]; and a contrasting between the essential stations for AFs (red dots) and those for DFs (blue dots) (c), where the thick black dots present their overlapped stations. (d) shows the cost-effective stations network including the**

**essential stations (blue, red, and black dots) as in (c) and the additional scattered stations (gray dots).**

Now we investigate how much this essential network can explain the skill improvement of the $PM_{2.5}$

forecasts when assimilating the data acquired from all the current ground meteorology stations in and

surrounding the BTH. As mentioned in section 3.2, we have to assimilate simulated observations taken

from the ERA5 due to the unavailable real observations. With the simulated observations, we assimilate

them from the essential stations and those from all the ground stations within and surrounding the BTH

to the control run generated by the GFS. Then comparisons between the assimilation runs and the control





runs can be made from the perspective of $AE_V$ and $AE_M$ in Eqs. (7) and (8), in attempt to show the role of the assimilated observations in improving PM$_{2.5}$ forecast skill.

$$AE_V = \left(\frac{|P_C-P_T|-|P_A-P_T|}{|P_C-P_T|}\right)_{t=T} \times 100\%, \quad (7)$$

$$AE_M = \frac{1}{T}\sum_{i=t_0}^{i=T}\left(\frac{|P_C-P_T|-|P_A-P_T|}{|P_C-P_T|}\right)_{t=i} \times 100\%, \quad (8)$$

where $AE_V$ and $AE_M$ measure the reduction rate of the errors in the control forecast at verification times [$T$, see Eq. (7)] and that during the whole forecast period [from $t_0$ to $T$, see Eq. (8)] after the assimilation. The $P_C$, $P_T$, and $P_A$ denote the surface PM$_{2.5}$ concentration in the control run, truth run and assimilation run, respectively. The sign $|\cdot|$ means the absolute value of forecast errors averaged over the BTH region.

For the 32 AFs, when assimilating the 99 simulated observations, the overall improvements are 12.03% and 13.59% measured by $AE_V$ and $AE_M$, respectively; and an average of 57% grids in the BTH area show positive $AE_V$ values and 54% grids show positive $AE_M$ values; particularly, the forecast with the largest forecast error among the 32 AFs presents a reduction rate of the error by 31.34% at the forecast time, even with approximately 76% of the grid points in the BTH area showing positive improvement (see Fig. 7 and Table 3). For the 16 DFs, assimilating the simulated observations at the 60 essential stations can improve the PM$_{2.5}$ forecast skills with the $AE_V$ varying from 4.12% to 45.53% (exactly from 0.57 to 15.18 µg m$^{-3}$) and the $AE_M$ varying from 0.03% to 39.24% (exactly from 0.34 to 7.77 µg m$^{-3}$) and the forecast errors are reduced by average for 18.07% at the forecast times and 18.05% during the whole forecast periods. It is indicated that, for either AFs or DFs, their respective essential stations can provide additional observations that much significantly increase the PM$_{2.5}$ forecasting skill in BTH region. Moreover, when the overall improvements are relative to those 15.48% and 17.90% (measured by $AE_V$ and $AE_M$) for AFs and 23.87% and 24.76% for DFs of assimilating the simulated observations taken from all the constructed stations within and surrounding the BTH (a total of 481 stations), they can account for at least 75% of the latter, although the former essential stations only cover at most 20.58% of the latter ground stations. It is clear that the essential stations can indeed provide additional observations that help increase the skill of the PM$_{2.5}$ forecast in the BTH much significantly, in comparison to other constructed stations but not in the sensitive grids. Therefore, the essential stations are indeed crucial for the improvement of the PM$_{2.5}$ forecasts in the BTH.



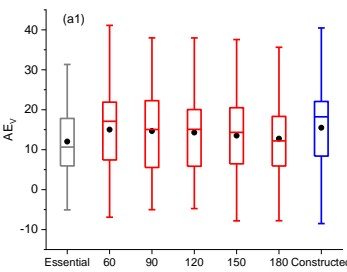 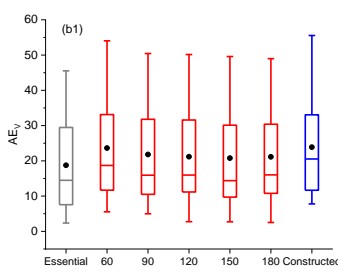


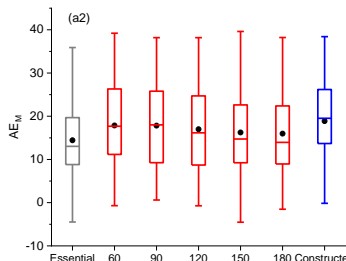 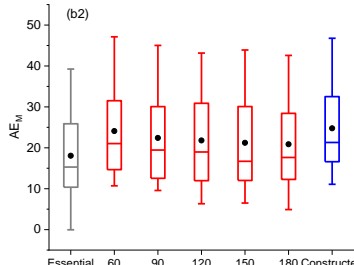

**Figure 7 The boxplot of the (1) $AE_V$ and (2) $AE_M$ values when assimilating the essential station observations, the essential observations plus the scattered station observations with the distances 60, 90, 120, 150 and 180km, and all constructed station observations for (a) AFs and (b) DFs.**

**4.2 The cost-effective observation network for significantly improving PM$_{2.5}$ forecast in the BTH**

The essential network has been shown to play the dominant role on the improvement of the PM$_{2.5}$ forecast in BTH when compared with assimilating the simulated observations taken from all constructed stations, but we notice that there still exist non-negligible differences between the improvements achieved by assimilating essential observations and those of assimilating all observations. Therefore, on

the current conditions of all the constructed stations, we would refine them to provide a cost-effective observation network that almost fully accounts for the total improvement of the PM$_{2.5}$ forecasts achieved by assimilating all the observations but brings fewer observations to the assimilation. For this purpose, we would base on the essential stations to further include relatively important stations from the remaining constructed ground stations (a total of 354 stations, which are defined by the exclusion of the 127

essential stations from the constructed 481 stations). For the remaining constructed stations, they are all also located on the areas covered by the CNOP-type errors for AFs and DFs but are ruled out of the first 3% grid points, therefore bearing very small errors. That is to say, the remaining stations are essential



neither for AFs nor for DFs and it is hard to distinguish whether they are more sensitive to AFs or DFs.

For example, the southwestern part of Shandong province is covered by some of the remaining stations,

but it not only locates on the area covered by the CNOP-type error of the AF initialized at the 14:00 on

Nov 2 in 2018, but also lies in the area of the CNOP-type error for the DF starting from the 14:00 on

Nov 4 in 2018 (see Fig. 5a, 5c). Therefore, to determine much useful ones of the remaining stations for

AFs and DFs, we do not distinguish which one is particularly important for AFs or DFs, but use the

comprehensive sensitivity ($r$TME) defined by Eq. (9) to balance its role on both AFs and DFs.

$$r\text{TME} = w_1 \frac{1}{n_1} \sum_{i=1}^{n_1} \text{TME}_{i(\text{AF})} + w_2 \frac{1}{n_2} \sum_{i=1}^{n_2} \text{TME}_{i(\text{DF})}, \quad (9)$$

where $\text{TME}_{i(\text{AF})}$ and $\text{TME}_{i(\text{DF})}$ represent the TME [see the Eq. (5)] of AF and DF, respectively. $n_1$

and  $n_2$  are the numbers of AFs and DFs, which are 32 and 16. Since the number of AFs is twice of DFs,

we define the weight coefficients $w_1 = \frac{1}{2}, w_2 = 1$. Thus, the sensitivity defined by the $r$TME could be

proportional to AF and DF and the grid points with larger $r$TME are expected to provide the additional

observations that, on the whole, contribute more to the reduction of the forecast errors for AFs and DFs.

Despite this, how many grid observations are needed to account for higher forecasting skills is also a

challenging problem, especially for those observations located on the non-sensitive grid points with small

CNOP-type errors. As shown by Liu and Rabier (2002), for a dense observation network with strongly

correlated error in the assimilation scheme, increasing the observation density may even decrease the

quality of analysis states and further decay the forecast skill. Particularly for the remaining ground

stations mentioned above, they locate the area covered by small errors in the CNOP-type error patterns

and are therefore less sensitive to $PM_{2.5}$ forecast uncertainties; then a worse forecast may come by when

the impacts of error correlations between the nearby observations overweigh the sensitivities. Therefore,

a decrease of observation density for the remaining stations is necessary to avoid the impairing analysis

in the assimilation process. In fact, Yang et al. (2014) suggested that assimilating the observations with

appropriate observing distance helps get larger benefits of the forecasts [see also Li et al., (2009), Zhang

et al., (2019), and Yang et al. (2022)]. Therefore, when we select relatively important stations from the

remaining stations by sorting the grid points according to the sensitivity provided by the $r$TME, we

should consider simultaneously the effect of station distances. To achieve it, we get an attempt to scatter

the remaining stations (354 in total) with the distances of 60, 90, 120, 150, 180 km and select the grid

point with much large values of $r$TME defined by Eq. (9) to determine the required stations. Note that





if the distance between the scattered stations is set smaller than 60km, all the remaining stations will be included which is inconsistent with the aim of refining. We take the scatter distance of 60km as an example to show how to select the required stations. Because the real station locations do not match the

grid points in the model, we take the $r$TME value of their closest grid point as an approximation of their sensitivities. Therefore, for the remaining constructed ground stations, the station whose closest grid point has the largest $r$TME is taken as the first selected ground station; then, we exclude the stations no further than 60km away from the first selected station and determine the station with the largest $r$TME among the rest stations as the second selected station; after the second station is determined, we further

exclude the stations no further than 60km away from the second station and selected the third station according to the $r$TME of its closest grid point; the other stations are similarly determined. Finally, a new observation network can be constructed by the combination of the essential stations and the scattered stations (see Fig. 6d).

The simulated observations (i.e. the ERA5 data) taken from the new observation networks are

assimilated to the control run to show the improvements achieved by assimilating the additional observations, where it is noted, since the essential stations responsible for DFs alone are not sensitive to the AFs, these stations are also scattered with corresponding distances according to the $r$TME when implementing the AFs; the same procedures are also carried out for the DFs. Specifically, on the basis of the essential stations, if the scattered stations are included with the distance of 60km, the performance of

the PM$_{2.5}$ forecasts for 32 AFs and 16 DFs are totally improved from 12.03% to 15.02% and from 18.07% to 23.62% measured by $AE_V$; meanwhile, the $AE_M$ increases from 13.59% to 17.15% and from 18.05% to 24.18% averaged by all the AFs and DFs, respectively (see Figure 7 and Table 3). If a comparison is made between the essential stations and the additional scattered stations, it is found that the latter contributes an improvement of 2.99% and 3.56% of the PM$_{2.5}$ forecasts measured by the $AE_V$ and $AE_M$

averaged for all the AFs and an improvement of 5.55% and 6.13% for all DFs, which, from another perspective, emphasizes the dominant role of the essential stations in improving the PM$_{2.5}$ forecasts. For the scattered stations with other distances above, we also do similar experiments and make comparisons with those scattered by the distance of 60km, eventually showing that the stations scattered by 60km perform the best in enhancing the PM$_{2.5}$ forecast skill for either AFs or DFs. However, we also find that

there are not big differences among the skill scores achieved by them. For example, when the additional



stations are scattered from 60km to 90km (correspondingly, the station number is further decreased by 83), the overall improvements of the AFs are only reduced by 0.35% measured by $AE_V$ and 0.02% measured by $AE_M$, while for the DFs, when the additional stations are scattered further than 90km, it is even difficult to differentiate the effects between the 120 to 180km distances. These imply that a

saturation of the error reduction may exist in the given framework. In fact, Morss et al. (2001) demonstrated that the analysis errors are often small in a certain density of observation network so that adding more observations only resulted in small benefits, which may explain the saturation of the error reduction in the $PM_{2.5}$ forecasts here.

Now we take the observation network constructed by the combination of essential stations and the
scattered stations with a distance of 60km as the newly refined observation network (see Figure 6d) and compare it with all the constructed ground stations by performing the assimilation runs. We find that the resultant improvements (15.02% for AFs and 23.62% for DFs; see above paragraph) by assimilating the newly refined station observations can account for 97% and 99% of the improvements (15.48% for AFs and 23.87% for DFs) achieved by assimilating all the constructed station observations for the AFs and

DFs, respectively. Particularly, among the individual forecasts, 9 of the 32 AFs and 5 of the 16 DFs even show much higher forecast skills at the forecast times in the assimilation of the newly refined observations than in that of all the constructed ground observations. It is demonstrated that assimilating the simulated observations on the refined network can result in comparative, sometimes even higher improvements of the $PM_{2.5}$ forecasting skills, as compared with assimilating all the ground stations

observations within and surrounding the BTH; furthermore, we note that the number of the newly refined stations is at least 180 less than that of the constructed stations. All these indicate that, on the condition of the current ground meteorological stations, the above newly refined stations may compose to a cost-effective observation network that almost accounts for the total improvement of the $PM_{2.5}$ forecasts achieved by assimilating all the ground observations. The cost-effective observation network may

provide guidance to optimize the current ground meteorological stations; at least, it suggests a much cost-effectively assimilation strategy for increasing the accuracy of meteorological forecasts for the significant improvement of the $PM_{2.5}$ forecasts in the BTH.

**Table 3 The mean and maxima of the improvements measured by $AE_V$/$AE_M$ for the AFs and DFs, when the**
**simulated observations on different observation networks are assimilated. The largest improvements among**



**AFs or DFs for the refined observation networks are marked in bold, respectively.**

| Observation network | AFs | | DFs | |
|---|---|---|---|---|
| | Mean (%) | Max (%) | Mean (%) | Max (%) |
| Essential stations | 12.03/13.59 | 31.34/35.89 | 18.07/18.05 | 45.53/39.24 |
| All constructed stations | 15.48/17.90 | 40.47/38.42 | 23.87/24.76 | 55.54/46.76 |
| Essential & Scattered 60km | **15.02/17.15** | **41.12/39.21** | **23.62/24.18** | **54.04/47.13** |
| Essential & Scattered 90km | 14.67/17.13 | 37.99/38.19 | 21.79/22.41 | 50.43/45.02 |
| Essential & Scattered 120km | 14.29/16.29 | 37.97/38.18 | 21.17/21.79 | 50.16/43.15 |
| Essential & Scattered 150km | 13.92/ 15.44 | 37.56/38.61 | 20.77/21.21 | 49.59/43.89 |
| Essential & Scattered 180km | 12.77/15.21 | 35.62/38.22 | 20.97/20.87 | 48.99/42.57 |

## 5. Interpretations

In this section, we interpret why assimilating the cost-effective station observations results in comparative improvements, sometimes even higher improvements in $PM_{2.5}$ forecasts than assimilating all the constructed station observations. It is known that the variation of $PM_{2.5}$ concentrations is dependent on both the thermodynamical and dynamic meteorological conditions. Beyond question, the stable thermodynamical conditions, such as low planetary boundary layer height, are favorable for the accumulation of the $PM_{2.5}$ concentrations (Miao et al., 2015); furthermore, a high relative humidity (RH) will also promote the processes such as heterogeneous chemistry and gas-particle partitioning, which are all favorable for the formation of the $PM_{2.5}$. For the dynamic conditions in BTH region, increased wind speed may conversely influence the $PM_{2.5}$ forecasts. For instance, dominant northerly wind will blow away the $PM_{2.5}$ in downtown areas of BTH region, whilst southerly wind will bring more $PM_{2.5}$ from the southern cities to the BTH region (Zhao et al., 2009). So the accuracy of thermodynamical and dynamic meteorological conditions are both essential for the $PM_{2.5}$ forecasts in the BTH region (see the review



paper of Chen et al., 2020).

        For all the AFs and DFs concerned in the study, we compare their meteorological conditions before
and after the assimilations of the cost-effective station observations and all the constructed station
observations, respectively. We find that the assimilation, as expected, adjusts the thermodynamical and
dynamic meteorological conditions at the initial state in the control run, and forecasts the meteorological

condition closer to the truth run which further improve the $PM_{2.5}$ forecasting skills. In particular, we
found that the improvements for the AFs are basically associated with the more accurate
thermodynamical conditions in the assimilation runs; whilst for the DFs, the improved forecasting skills
are mostly attributed to the corrections of both the dynamical and thermodynamical conditions.
Furthermore, the assimilations of the cost-effective station observations and all the constructed stations

observations correct the meteorological conditions for the $PM_{2.5}$ forecasts in a similar way, which thus
causes a comparative skill of the $PM_{2.5}$ forecasts between them. Specifically, we select two forecasts, i.e.
the AF initialized at the 14:00 on 2 November 2018 and the DF initialized at the 02:00 on 15 November
2018, which possess large forecast errors in the control runs, as examples to present the detailed
interpretations.

For the AF, the $PM_{2.5}$ concentrations in the truth run increases from 101.54 µg m$^{-3}$ at 14:00 on 2
November to 143.01 µg m$^{-3}$ at 02:00 on 3 November averaged over the BTH, indicating an
accumulation process of the $PM_{2.5}$. The control run is also able to present the accumulation process, but
with an underestimation of 129.92 µg m$^{-3}$ at the forecast time of 02:00 on 3 November (Fig. 8a). The
differences between them are mainly attributed to the thermodynamical condition, since there are less

differences in the wind components (see Fig. 9a). Therefore, we mainly concern the thermodynamical
condition to explain the AF. Compared with the truth run, the control run has presented a less stable
condition with an overestimation of 45.84 m in the boundary layer height and an underestimation of
16.67% in the RH averaged over the BTH region at the forecast time, which are not beneficial for the
accumulation and formation of $PM_{2.5}$ so that an underestimation of $PM_{2.5}$ concentration comes by. When

the simulated observations from all the constructed meteorological stations are assimilated, the boundary
layer height has decreased and the RH has increased over the central and southern part of BTH region at
the initial time. The improved thermodynamic condition further modifies the meteorological condition
at the forecast time, including a decrease of 21.56 m on the boundary layer height and an increase of





8.02% on RH averaged over the BTH region, which contribute to an increase of $PM_{2.5}$ concentrations from 129.92 μg m$^{-3}$ to 138.85 μg m$^{-3}$ averaged over the BTH region and thus an improvement of the $PM_{2.5}$ forecast skill (see Figure 10). By comparison, the assimilation of the cost-effective station observations will modify the meteorological conditions in the same way, with a decrease of 20.95 m in boundary layer and an increase of 7.57% in RH, finally resulting in the average $PM_{2.5}$ of 138.60 μg m$^{-3}$ at the forecast time, only 0.25 μg m$^{-3}$ lower than the forecast with the assimilation of all constructed station observations. Hence, the cost-effective network can approximate to the whole constructed stations and provide additional observations of equivalent efficiency to the whole observations in improving $PM_{2.5}$ forecasts in the BTH. Moreover, we also implement the $PM_{2.5}$ forecasts with longer lead times for the eight heavy haze events by using the meteorological analysis field updated by assimilating the cost-effective observations. And we demonstrate that, although the cost-effective network is developed according to the sensitivity on the meteorological forecasts with the lead time of 12 hours, its resultant meteorological analysis fields still have positive effects on improving the AFs with longer lead times. For example, in the AF quoted in this section, assimilating the cost-effective station observations can reduce the forecast errors by 32.05% and 7.81% at the forecast time with lead time of 18 and 24 hours, respectively; furthermore, these improvements are also approaching to those achieved by the assimilation of all the constructed station observations (see Figure 8a and Figure 10).

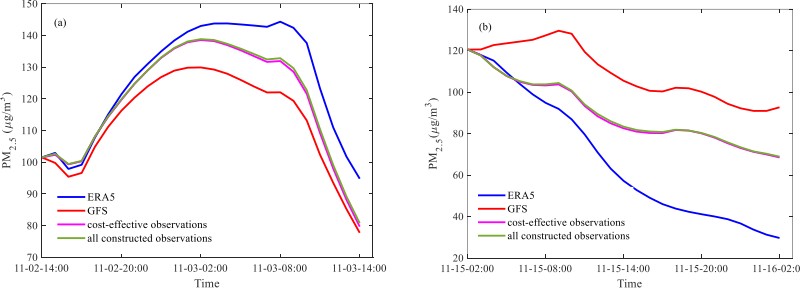

**Figure 8 Time series of the $PM_{2.5}$ concentrations averaged over the BTH region of the truth run, the control run, and the assimilation run inherited from the cost-effective observations and all the constructed observations for the AF initialized by 14:00 on 2 November 2018 with lead time of 24 hours (a) and the DF started from 02:00 on 15 November 2018 with lead time of 24 hours (b).**



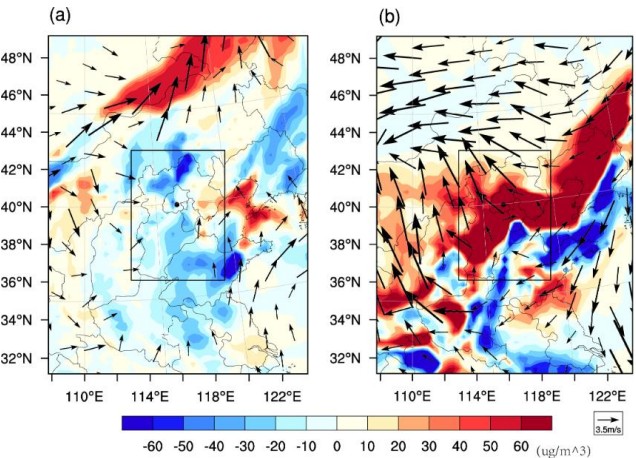

**Figure 9 The differences in the wind (vector, m s⁻¹) and PM₂.₅ concentration (shaded, ug m⁻³) between the**
**truth run and control run (control run minus truth run) for (a) the AF at the forecast time 02:00 on 3**
**November 2018 and (b) the DF at the forecast time 14:00 on 15 November 2018.**

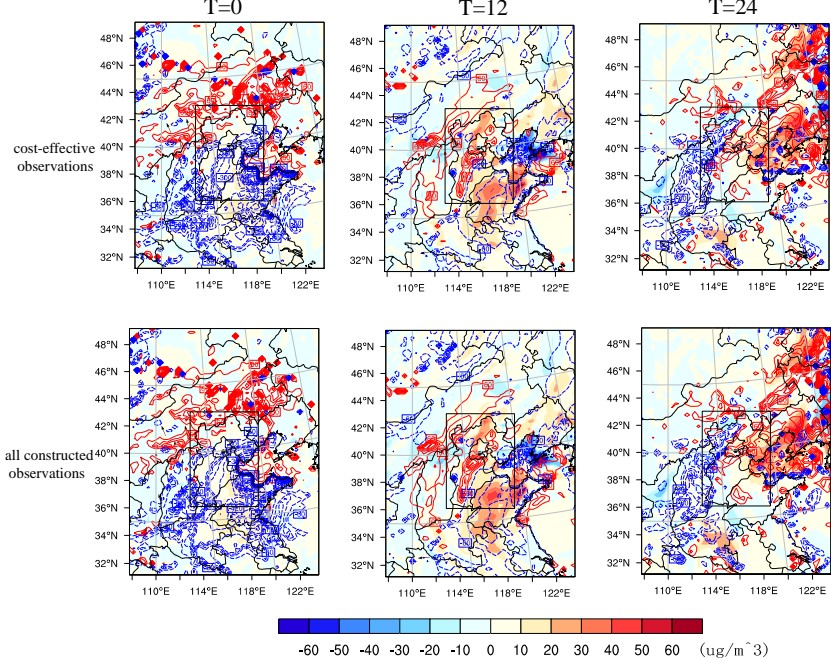

**Figure 10 The differences of the boundary layer height (contour line, m, blue line means reduction and red**
**line means increase) and the PM₂.₅ concentrations (shaded, μg m⁻³) between the assimilation run inherited**
**from the cost-effective observations and constructed observations and the control run for the AF started from**
**14:00 on 2 November 2018 and with the lead times of 12h and 24h.**

For the DF, the mechanism is different from the AF, where both thermodynamical and dynamical



conditions have critical impacts on the $PM_{2.5}$ variation. The $PM_{2.5}$ concentrations in the truth run decreased from 120.50 µg m$^{-3}$ at Nov 15$^{th}$ 02:00 to 57.19 µg m$^{-3}$ at Nov 15$^{th}$ 14:00 in the BTH region. The dissipation is caused by the northerly wind in the northwestern part of the BTH region at the initial

time, and then the northerly wind increased gradually with the speed of 4.92 m s$^{-1}$ at the forecast time over the BTH region, which blew away the $PM_{2.5}$ concentrations in the BTH. Conversely, the control run presents southerly wind in the northern part of BTH region and easterly wind in the Inner Mongolia Province, which are against the truth run (Figure 9b) and result in an overestimation of $PM_{2.5}$ with the concentration of 105.50 µg m$^{-3}$ at the forecast time (Figure 8b). Besides the dynamical reasons, the

control run also presents higher relative humidity biases over the BTH region, which also contributes to the overestimation of $PM_{2.5}$ concentrations. When the simulated observations from all the constructed stations are assimilated to the initial state, it increased the northwesterly wind in the northern part of the BTH region at the initial time and at the forecast time the northerly wind over the BTH region has increased to 2.73 m s$^{-1}$. Meanwhile, the assimilation also results in a decrease of RH from 76.28% to

73.67%. It is obvious that the increased northerly wind and decreased RH are beneficial for the dissipation of the $PM_{2.5}$ and lead to the $PM_{2.5}$ concentration decrease from 105.50 µg m$^{-3}$ to 83.35 µg m$^{-3}$ in the BTH region at the forecast time, resulting in an improvement of 45.85% $PM_{2.5}$ forecasting skills. When the simulated observations from the cost-effective station observations are assimilated, the meteorological conditions are modified in the same way, except the stronger northerly wind of 2.77 m s$^{-}$

$^{1}$ over the BTH at the forecast time. The stronger northerly wind blows more pollution in the BTH region to the downwind region so that the mean $PM_{2.5}$ concentrations over the BTH region decreases to 82.53 µg m$^{-3}$ and shows an improvement of 47.55% of $PM_{2.5}$ forecasting skills at the forecast time, 1.7% higher than the improvements when all the constructed station observations are assimilated. Therefore, though fewer observations in the cost-effective network are assimilated, they result in a higher

forecasting skill by reducing larger forecast errors in the northerly wind. Furthermore, similar to the AFs, with the meteorological analysis fields obtained by the cost-effect observation network, the comparative improvements of the DFs can be achieved at much longer times. Specifically in this forecast, the improvements can reach to 34.16% and 29.36% at the lead times of 18 and 24 hours measured by $AE_V$, respectively, almost the same as the improvements of the assimilation of all constructed station

observations (see Figure 8b and Figure 11).


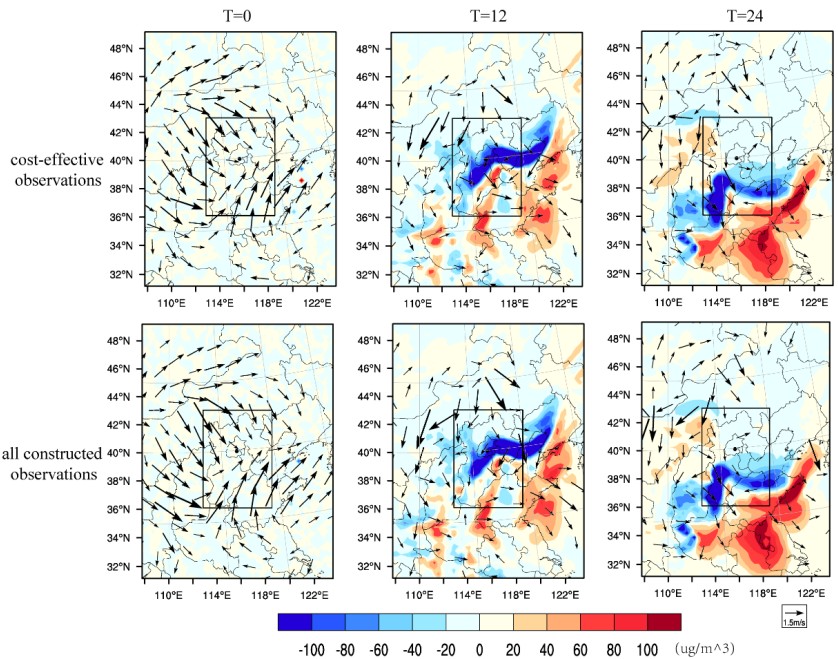

**Figure 11 The differences of ground wind (vector, m s$^{-1}$) and PM2.5 concentrations (shaded, μg m$^{-3}$) between the assimilation run (inherited from the cost-effective observations and all constructed stations observations) and the control run for the DF started from 02:00 on 15 November 2018 and with the lead times of 12h and 24 h.**

So far, we have verified numerically the validity of the cost-effective ground meteorological stations network in improving the PM$_{2.5}$ forecasts of the BTH more economically by assimilating fewer observations; also, we have interpreted this validity in terms of the perspective of dynamics and thermodynamics. It is therefore expected that the cost-effective network can provide a guidance to refine the current ground stations from the viewpoint of the PM$_{2.5}$ concentration forecasts in the BTH.

## 6. Summary and discussions

The PM$_{2.5}$ forecasts of BTH region are sensitive to meteorological initial condition and in this study, we investigate the role of the ground meteorological stations within and surrounding the BTH, finally proposing a strategy to refine them, inspired by the fact that high density of observations is not necessary to cause higher forecast benefits. Specifically, a total of 32 AFs and 16 DFs obtained from all eight heavy hazy events in the BTH region in the winter season during the years of 2016-2018 were investigated using the WRF-NAQPMS model; and their fastest growth initial errors, i.e. the CNOP-type errors, are



calculated to identify their respective sensitive areas of ground meteorological fields; based on these sensitive areas, a frequency method suggested by Duan et al. (2018) is used to recognize the sensitive

grid points applicable for the forecasts of the $PM_{2.5}$ concentrations with different start times, which provides help to refine the current ground meteorological stations (a total of 481 stations) within and surrounding the BTH and form a newly refined stations network (a total of 287 stations, which is 194 less than that of the former) for the $PM_{2.5}$ forecasts in the BTH.

Numerically, a series of OSSEs is conducted to verify the effectiveness of the newly refined 287

stations observations on improving the $PM_{2.5}$ forecasts in the BTH. They demonstrate that, when the additional simulated observations (i.e. the ERA5 data) from these refined stations are assimilated to the control run initialized by the GFS data the overall $PM_{2.5}$ forecasting skills increase to 15.02% and 23.62% at the forecast time of AFs and DFs, which have accounted for 97% and 99% of the improvements when the simulated observations from all the 481 ground stations are assimilated; especially, for some

individual forecasts, assimilating the simulated observations even results in higher forecasting skills of $PM_{2.5}$. Physically, we interpret why assimilating the fewer observations from the refined stations can have the improvement of the $PM_{2.5}$ forecast skill comparative to, even higher than that of assimilating the whole ground stations observations. In fact, assimilating the fewer observations has equivalent capabilities of correcting the atmospheric stability for the AFs and modifying the dynamical and

thermodynamical conditions for the DFs compared with assimilating the whole ground observations, which makes the control run closer to the truth and result in a comparative improvement of $PM_{2.5}$ forecast skills.

It is clear that assimilating the fewer observations can lead to higher $PM_{2.5}$ forecast skills, which indicates that it is not necessarily the use of much denser meteorological observation stations but instead

a few sensitive stations can greatly improve the $PM_{2.5}$ forecast skills. It implies that the 58% (the 279 refined stations for the AFs) of the current station observations accounting for the 97% of the improvements at the forecast time of AFs and 50% (the 241 refined stations for the DFs) of the current station observations contribute to the 99% of the improvements at the forecast time of DFs. Combined AFs and DFs, there are a total of 287 stations (about 60% of the current stations) remain to make highly

efficient contribution to the $PM_{2.5}$ forecasts in the BTH region. It is therefore indicated that the newly refined network may play a role of the cost-effective ground meteorological stations for greatly



improving the PM$_{2.5}$ forecast in the BTH. This may suggest that 287 refined stations in the study should maintain operations and other stations surrounding the BTH can be greatly scattered for avoiding the thankless work. Relative to the objective in scientifically arranging the observation network proposed by

the China Meteorological Administration during the 14[th] Five-Year Plan Period, our study would provide a scientific guidance for optimizing the ground meteorological station network with the respect of improving the air quality forecasts.

In this study, we focus on the effect of surface meteorological uncertainties of the PM$_{2.5}$ forecast in the BTH and suggest that the current constructed ground stations can be refined to a cost-effective station

network. In fact, these cost-effective stations, as demonstrated in Section 4, are made up of the constructed stations that are fallen into the area covered by the 174/184 sensitive grid points for AFs/DFs revealed by the CNOP-types errors and the scatted stations which have also been constructed but not fallen in the area covered by the sensitive grids. It is therefore conceivable that the cost-effective network in this study could be further optimized by moving the stations not located in the area covered by the

sensitive grids to the area with higher sensitivities (i.e. the area covered by the 174/184 sensitive grid points). Besides the meteorological observations, pollutant observations are also quite important for the air quality forecasts (Luo et al., 2022). Therefore, optimizing the environmental monitoring stations and obtaining more useful pollutant observations are also very important for the significant improvements of air quality forecasting, which may further reduce the gap between the forecasts and observations in the

air quality studies. Though previous studies have attempted to identify the sensitive areas for targeted observations of chemical constituents using singular vector or adjoint sensitivity methods (Daescu and Carmichael, 2003; Goris and Elbern, 2015), they used a linear approach and did not sufficiently consider the nonlinear effect of initial value sensitivity, so that implementing the observations on these sensitive areas may not lead to the largest improvements (Wang et al., 2011). The application of CNOP on

determining the sensitive areas may overcome the limitations. It is therefore expected that the optimization of environmental monitoring stations can be in-depth studied and more useful conclusions will be achieved for greatly improving the forecasts of air quality in the future.

**Code and data availability**

The WRF and its adjoint model used in this study is the version 3.6.1 and are available from the



website https://www2.mmm.ucar.edu/wrf/users/wrf_files/wrfv3.6/updates-3.6.1.html. The exact version of the model to produce the results used in this paper is available at Zenodo (https://doi.org/10.5281/zenodo.7627369, Yang and Duan, 2023). The analyzed data used in this paper is available at Zenodo (https://doi.org/10.5281/zenodo.7627556, Yang and Duan, 2023). Hourly surface PM$_{2.5}$ data are obtained from China National Environmental Monitoring Center (CNEMC, http://www.cnemc.cn/en/, CNEMC, 2022). The ERA5 reanalysis product is available at https://www.ecmwf.int/en/forecasts/datasets/reanalysis-datasets/era5 (Hersbach et al., 2017). The NCEP GFS product is available at https://rda.ucar.edu/datasets/ds084.1/ (NCEP, 2015).

**Author contributions**

YL and DW conceived the research, designed the experiments, performed the simulations and analyzed the results. All authors contributed to the drafting of the paper.

**Competing interests**

The contact author has declared that none of the authors has any competing interests.

**Acknowledgement**

The study was supported by the National Science Foundation of China (grant Nos. 42105061; 42142039).

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
