# Peer review of "An approach to refining the ground meteorological observation stations for improving $PM_{2.5}$ forecasts in Beijing-Tianjin-Hebei region"

_Geoscientific Model Development, 2023_

## Author Comment (AC1)

**Response to Reviewer #1**

We would like to thank the referee for reviewing the manuscript and providing the valuable comments and suggestions. We are sorry that for some sentences we did not make them clear in the manuscript. We will update our manuscript following the suggestions. Below we answer the specific comments point by point. For readability the comments are shown in bold and italics.

**General comments:**

*The authors try to refine the ground meteorological stations surrounding the Beijing-Tianjin-Hebei region to achieve an improved forecast for particulate matter. This topic is interesting and has practical implications since right now more and more stations are constructed but few studies have studied how they can help improve numerical forecasts in reality. The refining approach introduced in this paper by considering the sensitive areas is reasonable and logical. Overall, the manuscript is well written and clearly structured.*
*However, there are still several issues that should be addressed before acceptance.*

**Response:** We thank your appreciations.

**Major comments:**

1. *Based on the sensitive areas identified by the CNOP associated with the 48 forecasts, the authors first identify the essential observation network, and then scatter the remaining station according to the comprehensive sensitivity. So the accurate calculate of CNOP is the basis of the study. In section 2.3, the authors presented a detailed description on the definition of CNOP-type error. However, the descriptions on how to calculate the CNOP is brief and insufficient. I suggest the authors add more details of the algorithm on Line 195.*

**Response:** We are sorry that we do not present much sufficient information on the algorithm of the CNOP-type errors. We will add the following details on Line 195 in the revised manuscript.

"The spectral projected gradient 2 (SPG2) method is used to solve the optimization problem in Eq. (3). It is noted that the SPG2 algorithm is generally designed to solve the minimum value of nonlinear function (cost function) with an initial constraint condition, and the gradient of cost function with respect to the initial perturbation represents the descending direction of searching for the minimum of the cost function. Therefore, in this study, we have to rewrite the cost function Eq.(3) as $J'(\delta x_0^*) = \min_{\delta x_0{}^T C_1 \delta x_0 \leq \beta} - [M(x_0 + \delta x_0) - M(x_0)]^T C_2 [M(x_0 + \delta x_0) - M(x_0)]$ and the WRF adjoint model is used to compute the gradient of the cost function. Specially, to calculate the CNOP, a first guess initial perturbation is projected into the constraint condition $(\delta x_0^{(0)})$ and superimposed on the initial state $(x_0)$ of the WRF model. After the forward

integration of WRF, the value of cost function, $-[M(x_0 + \delta x_0^{(0)}) - M(x_0)]$, can be obtained. Then, with the adjoint model of WRF, the gradient of the cost function with respect to the initial perturbation $(g(\delta x_0^{(0)}))$ is calculated. Ideally, the gradient presents the fastest descending direction of the cost function. However, in realistic numerical experiments, the gradient presents the fast-descending direction but not necessarily the fastest, so we need many more times of iterations. After iteratively forward and backward integrations of the WRF model governed by SPG2 algorithm, the initial perturbation is optimized and updated until the convergence condition is satisfied. Here, the convergence condition is $\left\|P(\delta x_0^{(p)} - g(\delta x_0^{(p)})) - \delta x_0^{(p)}\right\|_2 \le \varepsilon_1$, where $\varepsilon_1$ is an extremely small positive number, $P(\delta x_0^{(p)})$ projects the initial perturbation to the constraint condition. Finally, the CNOP $(\delta x_0^{(p)})$ which presents the initial perturbation that causes the largest forecast errors using the SPG2 method can be obtained. To make it clearer, we add a flow chart of the CNOP calculation in the revised manuscript.

[Figure]

Figure 1 the flow chart of CNOP calculation

2. *In section 5. The authors select two forecasts which possess large forecast errors in the control run as examples to show that the cost-effective stations provide observations of equivalent efficiency of the whole constructed stations. However, to better demonstrate the effectiveness of the cost-effective observations, I recommend the authors to have a look at the improvements when the cost-effective observations are removed from the whole station observations. If the remained observations (after the removal of the cost-effective observations) contribute to a slight improvement of the PM2.5 forecasts but with a larger number, then it will be more convincing that the cost-effective observations are necessary for the PM2.5 forecasts in BTH.*

    **Response:** We thank the reviewer's suggestions. The CNOP-type error represents

the initial error that results in the largest forecast error in the verification area at the verification time. The CNOP-type error considers the interaction among the errors on spatial grid points and in this situation, the errors on the grid points with large amplitude of the CNOP-type error contribute much more to the final prediction error. When we sort the spatial grid points with a decreasing order according to the amplitude of the error and choose the first 3% grid points as the essential grid points, the interactions between these grid points are remained, so that it is assumed that assimilating the observations on these grid points may contribute more to the improvements of forecast skills. Based on a series of OSSEs, it is verified that assimilating the essential or cost-effective observations can indeed improve greatly the PM2.5 forecasts. Specifically, when the 279 cost-effective station observations are assimilated for the AFs, they achieve an overall 41.11% the improvement of PM2.5 forecasting skills, which explains 99% the improvement when assimilating constructed station observations; furthermore, when the cost-effective station observations are removed from all the constructed station observations, the number of the rest station observations is 77 smaller than that of the cost-effective station observations and the assimilation of these observations explains much less, which is 70% the improvement obtained by assimilating all constructed station observations. To be specially emphasized, for the DFs, when the simulated observations from the 241 cost-effective station observations are assimilated, it results in an improvement of 47.55% of PM2.5 forecasting skills, even 1.7% higher than the improvement of assimilating all constructed station observations; however, when the cost-effective station observations are removed, assimilating the rest 240 station observations would only result in an improvement of 22.60% PM2.5 forecasting skill. Obviously, although the number of rest station observations is almost the same with the cost-effective station observations, the improvement of PM2.5 forecasting skills is less than half of the improvements obtained by assimilating the cost-effective station observations.

Totally, assimilating the cost-effective station observation will lead to much higher PM2.5 forecasting skills than assimilating the rest observations, which emphasizes the important role of the cost-effective station observations in improving the PM2.5 forecast skills. The relevant results and discussions will be added in the revised manuscript.

3. *Also, in section 5, the authors only take two examples to present the detailed interpretations, which is not enough to me. Even if the authors have explained on Line 584 that the assimilations of the cost-effective station observations and all the constructed station observations correct the meteorological conditions for the PM2.5 forecasts in a similar way, it is suggested to add more examples or discuss the overall corrected meteorological conditions in more detail. For example, the authors may use the atmospheric stability to quantify the meteorological conditions for the accumulation or dissipation of PM2.5 concentrations.*

**Response:** We thank the reviewer's suggestions. For all the AFs and DFs in the study, we have compared their meteorological conditions before and after the assimilations of the cost-effective station observations and all the constructed station observations, respectively. We find that for the AFs, assimilating the cost-effective station observations will adjust the atmospheric stability; and for the DFs, assimilating the cost-effective observations will correct both the dynamical and thermodynamical meteorological conditions, as we discussed on Lines 576-585 in the manuscript. Specially, we select two forecasts as examples to show the details. The other forecasts show similarities with the two example forecasts that assimilating the cost-effective station observations and all the constructed station observations correct the meteorological conditions in a similar way, which causes a comparative skill of PM2.5 forecasts. To make the interpretations clear and not superfluous, we think the interpretations in the present manuscript are acceptable; if more examples are included, it is much difficult to make the content logical.

**Minor comments:**

1. *Line 104. For the application of CNOP in field campaigns, Feng et al., (2022) demonstrated its validity on identifying sensitive areas for typhoon forecasting.*
*Feng, J., Qin, X., Wu, C., and coauthors. Improving typhoon predictions by assimilating the retrieval of atmospheric temperature profiles from the FengYun-4A's Geostationary Interferometric Infrared Sounder (GIIRS). Atmospheric Research, 280(15), 106391.*
**Response:** We thank the reviewer to providing the reference. We have read the paper and will cite it in the manuscript.

2. *Line 288. The authors use "target observation" here, but in the introduction part they used "targeted observation". Please unify the usage.*
**Response:** We will modify the "target observation" to "targeted observation" on Line 288. We will also check its usages throughout the paper.

3. *Line 290. When the "cost-effective" first appeared in the manuscript, I did not quite understand what it means. More explanations should be added here.*
**Response:** Sorry for confusing the reviewer. The "cost-effective" means assimilating the observations obtained from fewer meteorological stations could lead to higher PM2.5 forecasting skills. This kind of station network can be taken as cost-effective stations because it provides sensitive observations to the PM2.5 forecasts in the economic fashion. The explanations will be added in the revised manuscript.

4. *Line 323, when determining the sensitive areas, the authors should clarify here that CNOP-type initial errors are superimposed on the ground meteorological fields in the "truth run".*
**Response:** The CNOP-type initial errors superimposed on the ground meteorological fields are calculated for each of the 48 PM2.5 forecasts in the "truth run" with the application of WRF and its adjoint model by using the SPG2 solver (see section 2). We

will rephrase the sentence in the revised manuscript.

**5. *Line 420, the 110ºE~120E should be 110ºE~120º Also the 34N~36N.***
**Response:** We will correct "110E~120E" to "110ºE~120ºE". We will also correct "34N~36N" to "34ºN~36ºN".

**6. *Line 705, it is recommended to mention in the section 6 that the improvements are based on the OSSEs, which means the simulated observations from ERA5 are assimilated to the control run to show the effectiveness of the newly refined station observations. However, how the improvements will be when the real observations from the refined station network are assimilated still needs further studies.***
**Response:** We thank the reviewer's suggestion. We will add discussions in the revised manuscript.

As we showed on Lines 295-315 in the manuscript, to identify the sensitive area of the ground meteorological field in each forecast, we adopt the idea of Lorenz (1965) and take the better simulation initialized by ERA5 as "truth run" and the simulation initialized by GFS forecast data as "control run", where these two simulations have the same emission inventory and use the same model; so the difference between them reflect the sensitivities of forecast uncertainties of PM2.5 concentrations on the accuracy of initial meteorological field. When we compute the CNOP-type initial perturbation superimposed on the better simulation initialized by ERA5, it can be regarded as an approximation to the most sensitive initial error and the sensitive area identified by such CNOP-type error can be regarded as an approximation to the real sensitive area. If the approximate sensitive area is valid, assimilating the additional observations in the sensitive area of control forecast will make the updated forecasts approach to the truth run.

Although the present study is associated with hindcasts of PM2.5, it is still difficult to obtain the meteorological observations from the Monitor Center; therefore, we can only assimilate the simulated observations (i.e. the ERA5 data) to the control run to show the effectiveness of the cost-effective observation network. If the cost-effective station network is useful along this thinking, it can be inferred that assimilating real observations from the cost-effective stations to the initial field of the meteorological of the control forecast would improve the meteorological field forecasting and then the PM2.5 forecasting greatly against the observations.

**7. *The boundary layer height is also an important meteorological variable for PM2.5 forecasts. Why do not the authors consider the perturbation of this variable in the study?***
**Response:** The CNOP in the present study only considers the sensitivity from initial uncertainties. We agree with the reviewer that the boundary layer height is an important meteorological variable for PM2.5 forecasts. Since the boundary layer simulation is more influenced by the parameterization in the WRF model (Chen et al., 2017; Mohan and Gupta, 2018), to study the role of boundary layer uncertainties in yielding the

PM2.5 forecast uncertainties, an extension of the CNOP method, CNOP-parametric perturbation (CNOP-P; Mu et al., 2010) or nonlinear forcing singular vector (NFSV, Duan and Zhou, 2013), can be used to identify the sensitivities of boundary layer uncertainties. The related discussions will be added in the revised manuscript.

**References:**

Chen, D., Xie, X., Zhou, Y., Lang, J., Xu, T., Yang, N., Zhao, Y., Liu, X., 2017. Performance evaluation of the wrf-chem model with different physical parameterization schemes during an extremely high PM2.5 pollution episode in Beijing. Aerosol Air Qual. Res. 17 (1), 262–277.

Duan, W., and Zhou, F., 2013. Non-linear forcing singular vector of a two-dimensional quasi-geostrophic model. Tellus, 65(18452), 256-256.

Mohan, M. and Gupta, M., 2018. Sensitivity of PBL parameterizations on PM10 and ozone simulation using chemical transport model WRF-Chem over a sub-tropical urban airshed in India. Atmospheric Environment, 185, 53-63.

Mu, M., Duan, W. S., Wang, Q., and Zhang, R., 2010. An extension of conditional nonlinear optimal perturbation approach and its applications, Nonlin. Processes Geophys., 17(2), 211-220.

---

## Author Comment (AC2)

**Response to Reviewer #2:**

The manuscript entitled "An approach to refining the ground meteorological observation stations for improving PM2.5 forecasts in Beijing-Tianjin-Hebei region" introduced an approach to refine the ground stations by identifying the sensitive areas for targeted observations. The study is highly related to the studies of predictability, target observation and data assimilation. And it provides a scientific guidance on optimizing the ground stations. I believe the approach is not only useful for air quality forecasts, but can also be used to the forecasts of extreme weather events. Nevertheless, there is a gap between publication and the manuscript in current version. I hope the following comments will help authors improve the manuscript.

**Response:** We thank your appreciations.

**Specific comments:**
*1. Line 42. There are a great many publications addressing the meteorological conditions on PM2.5 variations, but the authors only cite one, which is not enough. More references are needed here.*
**Response:** We thank the reviewer's suggestions. We will add the references on Line 42 (Lou et al., 2019; Chen et al., 2020).

*2. Line 68. "assimilating more observations may not necessarily lead to much higher forecast benefits." References are needed here.*
**Response:** We will add the references here (Li et al., 2010; Liu et al., 2021).

*3. Line 75. How are the worse forecast skills possible when the sensitivities are low? Please provide a detailed explanation here.*
**Response:** We will add a detailed explanation in the revised manuscript. Theoretically, if the observations in the area where the forecast is not sensitive to the initial errors are assimilated, the forecast skills might be slightly improved or neutral. However, in realistic forecasts, the imperfect assimilation procedure or the unresolved scales and processes in the model may induce additional errors and lead to the worse forecasts when the observations in the area where the forecast is not sensitive to the initial errors are assimilated (Janjic et al., 2018). For example, in Yu et al. (2012), removing the initial error in the area that is not the most sensitive area could worsen the prediction results of ENSO. That emphasized the importance of identifying the most sensitive area and suggests that additional observations should be assimilated more carefully in this sense.

*4. Line 195-202. The descriptions are insufficient and confuse me. Please add more details and make it clear.*
**Response:** Sorry for confusing the reviewer. We will rewrite the paragraph and make it clearer.
"The spectral projected gradient 2 (SPG2) method is used to solve the

optimization problem in Eq. (3). It is noted that the SPG2 algorithm is generally designed to solve the minimum value of nonlinear function (cost function) with an initial constraint condition, and the gradient of cost function with respect to the initial perturbation represents the descending direction of searching for the minimum of the cost function. Therefore, in this study, we have to rewrite the cost function Eq.(3) as $J'(\delta x_0^*) = \min\limits_{\delta x_0^T C_1 \delta x_0 \leq \beta} -[M(x_0 + \delta x_0) - M(x_0)]^T C_2[M(x_0 + \delta x_0) - M(x_0)]$ and the WRF adjoint model is used to compute the gradient of the cost function. Specially, to calculate the CNOP, a first guess initial perturbation is projected into the constraint condition $(\delta x_0^{(0)})$ and superimposed on the initial state $(x_0)$ of the WRF model. After the forward integration of WRF, the value of cost function, $-[M(x_0 + \delta x_0^{(0)}) - M(x_0)]$, can be obtained. Then, with the adjoint model of WRF, the gradient of the cost function with respect to the initial perturbation $(g(\delta x_0^{(0)}))$ is calculated. Ideally, the gradient presents the fastest descending direction of the cost function. However, in realistic numerical experiments, the gradient presents the fast-descending direction but not necessarily the fastest. So we need many more times of integrations. After iteratively forward and backward integrations of the WRF model governed by SPG2 algorithm, the initial perturbation is optimized and updated until the convergence condition is satisfied. Here, the convergence condition is $\left\| P(\delta x_0^{(p)} - g(\delta x_0^{(p)})) - \delta x_0^{(p)} \right\|_2 \leq \varepsilon_1$, where $\varepsilon_1$ is an extremely small positive number, $P(\delta x_0^{(p)})$ projects the initial perturbation to the constraint condition. Finally, the CNOP $(\delta x_0^{(p)})$ which presents the initial perturbation that causes the largest forecast errors using the SPG2 method can be obtained.

*5. Line 313. Please clarify that the real "meteorological" observations are not in public archive, because in section 3.1, the authors have compared the simulations with the observed PM2.5 concentrations.*

**Response:** We will clarify that the real meteorological observations are not available in public archive in the revised manuscript. The sentence will be corrected into "Since the real meteorological observations are not in public archive, the "additional observations" are correspondingly taken from the initial field of the truth run (i.e. the ERA5 data) and called as "simulated observations" according to the OSSEs.".

*6. Line 322. Is the CNOP-type initial error that what has been described in section 2.3? It is suggested to add a detailed description on what variables the CNOP-type errors have contained here.*

**Response:** Yes, the CNOP-type initial error is what has been described in section

2.3. We will add a detailed description of CNOP-type error here. The sentence will be corrected into "the CNOP-type initial errors which includes wind, temperature and water vapor mixing ratio components at the ground level are calculated for each of the 48 PM2.5 forecasts with the application of WRF and its adjoint model by using the SPG2 solver (see section 2)."

**7. Line 349. "the area with larger values of TME can be regarded as the sensitive areas". It is ambiguous. Is there a threshold for the definition of sensitive area or just determined subjectively?**

**Response:** The TME is applied to measure the comprehensive sensitivity of PM2.5 forecast uncertainties on initial meteorological perturbations. When we identify the essential observational network, we take the 3% as the threshold to determine the sensitive area. Then a total of 424 sensitive grid points is obtained. We select the 3% as the threshold here because the number 424 of sensitive grid points is close to the number of 481 of the meteorological stations within and surrounding the BTH region.

**8. Line 453. "the essential stations can indeed provide additional observations that help increase the skill of the PM2.5 forecasts, in comparison to other constructed stations but not in the sensitive grids". The authors did not do any comparison experiments to show the improvements are higher than assimilating the station observations which are not in the sensitive grids. How can they get such conclusions?**

**Response:** We will correct the sentence into "It is clear that the essential stations can indeed provide additional observations that help increase the skill of the PM2.5 forecast in the BTH much significantly".

**9. As shown in Figure 7 (a1, a2), assimilating the observations will lead to worse forecasts since the AEv and AEM are negative. It is hard to understand. Why will assimilating the observations will lead to worse forecasts?**

**Response:** The negative PM2.5 forecast skills occurred at the AF initialized at 20:00 on Nov 18[th] 2016. For the AF initialized at 20:00 at Nov 18[th] 2016, the PM2.5 concentration in the truth run increases from 139.5 $\mu g\,m^{-3}$ to 151.5 $\mu g\,m^{-3}$ averaged over the BTH region; while the control run forecasts the PM2.5 concentration of 159.6 $\mu g\,m^{-3}$ averaged over the BTH region, 20.1 $\mu g\,m^{-3}$ higher than the PM2.5 concentration in the truth run. When all the constructed station observations are assimilated, the PM2.5 concentration averaged over the BTH region is 153.11 $\mu g\,m^{-3}$ at the forecast time, much closer to the truth run. However, the improvements in the BTH region are uneven (Figure 2), and the number of grids showing negative improvements overweigh those showing positive improvements, which results in a negative AEv and $AE_M$.

[Figure]

Figure 2 The improvements of PM2.5 forecast skills when all the constructed station observations are assimilated.

*10. Line 703. "It is clear that assimilating the fewer observations can lead to higher PM2.5 forecast skills". It is inaccurate. It is suggested to rephrase it more carefully.*

**Response:** We will correct the sentence into "It is clear that assimilating the fewer sensitive observations may lead to higher PM2.5 forecast skill".

*11. It is suggested to mention the limitation of the study in section 6 that the results are based on OSSEs. If the real observations are available, how the refined station observations help improve the air quality forecasts deserve deeper studies.*

**Response:** We will add the limitations in the revised manuscript. Due to the unavailable of the meteorological observations from the Monitor center, we have to assimilate the simulated observations (the ERA5 data) to the control run to show the effectiveness of the cost-effective observation network. If the real meteorological observations are available, how the real observations from the refined station network can help improve the PM2.5 forecasts in the control forecast against the observations still needs further studies.

**References:**
Janjić, T, Bormann, N, Bocquet, M, Carton, JA, Cohn, SE, Dance, SL, Losa, SN, Nichols, NK, Potthast, R, Waller, JA, Weston, P., 2018. On the representation error in data assimilation, Q J R Meteorol Soc. 144: 1257– 1278.

Li, X., Zhu, J., Xiao, Y., and Wang, R., 2010. A Model-Based Observation-Thinning Scheme for the Assimilation of High-Resolution SST in the Shelf and Coastal Seas around China, Journal of Atmospheric and Oceanic Technology, 27(6), 1044-1058.

Liu C., Zhang, S., Gao, Y., Wang, Y. and coauthors, 2021. Optimal estimation of initial concentrations and emission sources with 4D-Var for air pollution prediction in a 2D

transport model. Science of the Total Environment, 773, 145580.

Lou, M., Guo, J., Wang, L., Xu, H., Chen, D., Miao, Y., Lv, Y., Li, Y., Guo, X., Ma, S., Li, J., 2019. On the relationship between aerosol and boundary layer height in summer in China under different thermodynamic conditions. Earth Space Sci. 6 (5), 887–901.

Yu, Y., Mu, M., Duan, W., Gong, T., 2012. Contribution of the location and spatial pattern of initial error to uncertainties in El Niño predictions. Journal of Geophysical Research, 117, C06018.

---

## Author Response (AR2)

**Response to Reviewers:**

We highly appreciate the two anonymous reviewers and one community reviewer who provided constructive comments, that greatly improved the overall quality of the paper. For the Response to Reviewer #2, please turn to Page 7. For the Response to Community Comment, please turn to Page 12. The line numbers refer to the track changes version.

**Response to Reviewer #1**

We would like to thank the referee for reviewing the manuscript and providing the valuable comments and suggestions. We are sorry that for some sentences we did not make them clear in the manuscript. We have updated our manuscript following the suggestions. Below we answer the specific comments point by point. For readability the comments are shown in bold and italics.

**General comments:**

***The authors try to refine the ground meteorological stations surrounding the Beijing-Tianjin-Hebei region to achieve an improved forecast for particulate matter. This topic is interesting and has practical implications since right now more and more stations are constructed but few studies have studied how they can help improve numerical forecasts in reality. The refining approach introduced in this paper by considering the sensitive areas is reasonable and logical. Overall, the manuscript is well written and clearly structured.***
***However, there are still several issues that should be addressed before acceptance.***

**Response:** We thank your appreciations.

**Major comments:**

1. ***Based on the sensitive areas identified by the CNOP associated with the 48 forecasts, the authors first identify the essential observation network, and then scatter the remaining station according to the comprehensive sensitivity. So the accurate calculate of CNOP is the basis of the study. In section 2.3, the authors presented a detailed description on the definition of CNOP-type error. However, the descriptions on how to calculate the CNOP is brief and insufficient. I suggest the authors add more details of the algorithm on Line 195.***
**Response:** We are sorry that we do not present much sufficient information on the algorithm of the CNOP-type errors. We have added the following details in the revised manuscript. **Please see Lines 205-219.**
 "The spectral projected gradient 2 (SPG2) method is used to solve the optimization problem in Eq. (3). It is noted that the SPG2 algorithm is generally designed to solve

the minimum value of nonlinear function (cost function) with an initial constraint condition, and the gradient of cost function with respect to the initial perturbation represents the descending direction of searching for the minimum of the cost function. Therefore, in this study, we have to rewrite the cost function Eq.(3) as $J'(\delta x_0^*) = \min\limits_{\delta x_0^T C_1 \delta x_0 \leq \beta} -[M(x_0 + \delta x_0) - M(x_0)]^T C_2 [M(x_0 + \delta x_0) - M(x_0)]$ and the WRF adjoint model is used to compute the gradient of the cost function. Specially, to calculate the CNOP, a first guess initial perturbation, $\delta x_0^{(0)}$, is projected into the constraint condition and superimposed on the initial state $x_0$ of the WRF model. After a forward integration of the WRF, the value of the cost function, i.e. $-[M(x_0 + \delta x_0^{(0)}) - M(x_0)]$, can be obtained. Then, with the adjoint model of the WRF, the gradient of the cost function with respect to the initial perturbation, $g(\delta x_0^{(0)})$, is calculated. Theoretically, the gradient presents the fastest descending direction of the cost function. However, in realistic numerical experiments, the gradient presents the fast-descending direction but not necessarily the fastest, so we need more iterations. After iteratively forward and backward integrations of the WRF model governed by the SPG2 algorithm, the initial perturbation is optimized and updated until the convergence condition is satisfied, where the convergence condition is $\left\| P\left(\delta x_0^{(p)} - g\left(\delta x_0^{(p)}\right)\right) - \delta x_0^{(p)} \right\|_2 \leq \varepsilon_1$ and

$\varepsilon_1$ is an extremely small positive number, $P(\delta x_0^{(p)})$ projects the initial perturbation to the constraint condition. Finally, the CNOP $\delta x_0^{(p)}$ can be obtained. The flow chart of the CNOP calculation is shown in Figure 2. For further details of the SPG2 algorithm, the readers can be referred to Birgin et al. (2001). To make it clearer, we add a flow chart of the CNOP calculation in the revised manuscript.

[Figure]

Figure 1 the flow chart of CNOP calculation

*2. In section 5. The authors select two forecasts which possess large forecast errors in the control run as examples to show that the cost-effective stations provide observations of equivalent efficiency of the whole constructed stations. However, to better demonstrate the effectiveness of the cost-effective observations, I recommend the authors to have a look at the improvements when the cost-effective observations are removed from the whole station observations. If the remained observations (after the removal of the cost-effective observations) contribute to a slight improvement of the PM2.5 forecasts but with a larger number, then it will be more convincing that the cost-effective observations are necessary for the PM2.5 forecasts in BTH.*

**Response:** We thank the reviewer's suggestions. The CNOP-type error represents the initial error that results in the largest forecast error in the verification area at the verification time. The CNOP-type error considers the interaction among the errors on spatial grid points and in this situation, the errors on the grid points with large amplitude of the CNOP-type error contribute much more to the final prediction error. When we sort the spatial grid points with a decreasing order according to the amplitude of the error and choose the first 3% grid points as the essential grid points, the interactions between these grid points are remained, so that it is assumed that assimilating the observations on these grid points may contribute more to the improvements of forecast skills. Based on a series of OSSEs, it is verified that assimilating the essential or cost-effective observations can indeed improve greatly the PM2.5 forecasts. Specifically, when the 279 cost-effective station observations are assimilated for the AF in section 5, they achieve an overall 41.11% the improvement of PM2.5 forecasting skills, which explains 99% the improvement when assimilating constructed station observations; furthermore, when the cost-effective station observations are removed from all the constructed station observations, the number of the rest station observations is 77 smaller than that of the cost-effective station observations and the assimilation of these observations explains much less, which is 70% the improvement obtained by assimilating all constructed station observations. To be specially emphasized, for the DF presented in section 5, when the simulated observations from the 241 cost-effective station observations are assimilated, it results in an improvement of 47.55% of PM2.5 forecasting skills, even 1.7% higher than the improvement of assimilating all constructed station observations; however, when the cost-effective station observations are removed, assimilating the rest 240 station observations would only result in an improvement of 22.60% PM2.5 forecasting skill. Obviously, although the number of rest station observations is almost the same with the cost-effective station observations, the improvement of PM2.5 forecasting skills is less than half of the improvements obtained by assimilating the cost-effective station observations.

Totally, assimilating the cost-effective station observation will lead to much higher PM2.5 forecasting skills than assimilating the rest observations, which emphasizes the important role of the cost-effective station observations in improving the PM2.5 forecast skills.

In the current version of manuscript, we have considered the interactions among

the errors on spatial grid points so that we choose the grid points with large amplitude of the CNOP-type errors, which will have large impact on final prediction result, to construct the cost-effective observation network. So the cost-effective observation network is constructed under the consideration of the interactions among the errors on spatial grid points. Moreover, it is verified numerically that assimilating these cost-effective station observations can achieve the PM2.5 forecasting skills comparable to that obtained by assimilating all ground station observations. We think it is enough to demonstrate the effectiveness of cost-effective station observations. However, if we assimilate the rest station observations, since the interactions among errors on spatial grid points are considered, the impact of the cost-effective station observations could also be reflected in the improvement of forecasting skill obtained by the assimilation of rest station observations. That's why assimilating the rest station observations could explain roughly 70% the improvement obtained by assimilating all constructed station observations in the AF shown in section 5. In this situation, it is hard to distinguish the role of cost-effective station observation. Therefore, even though we have implemented the experiments suggested by the reviewer, to avoid confusing the readers, we think it is better to keep the original structure and do not include the additional experimental results in the revised manuscript.

3. ***Also, in section 5, the authors only take two examples to present the detailed interpretations, which is not enough to me. Even if the authors have explained on Line 584 that the assimilations of the cost-effective station observations and all the constructed station observations correct the meteorological conditions for the PM2.5 forecasts in a similar way, it is suggested to add more examples or discuss the overall corrected meteorological conditions in more detail. For example, the authors may use the atmospheric stability to quantify the meteorological conditions for the accumulation or dissipation of PM2.5 concentrations.***

**Response:** We thank the reviewer's suggestions. For all the AFs and DFs in the study, we have compared their meteorological conditions before and after the assimilations of the cost-effective station observations and all the constructed station observations, respectively. We find that for the AFs, assimilating the cost-effective station observations will adjust the atmospheric stability; and for the DFs, assimilating the cost-effective observations will correct both the dynamical and thermodynamical meteorological conditions, as we discussed on Lines 576-585 in the manuscript. Specially, we select two forecasts as examples to show the details. The other forecasts show similarities with the two example forecasts that assimilating the cost-effective station observations and all the constructed station observations correct the meteorological conditions in a similar way, which causes a comparative skill of PM2.5 forecasts. To make the interpretations clear and not superfluous, we think the interpretations in the present manuscript are acceptable; if more examples are included, it is much difficult to make the content logical.

**Minor comments:**

1. *Line 104. For the application of CNOP in field campaigns, Feng et al., (2022) demonstrated its validity on identifying sensitive areas for typhoon forecasting.*
*Feng, J., Qin, X., Wu, C., and coauthors. Improving typhoon predictions by assimilating the retrieval of atmospheric temperature profiles from the FengYun-4A's Geostationary Interferometric Infrared Sounder (GIIRS). Atmospheric Research, 280(15), 106391.*
**Response:** We thank the reviewer to providing the reference. We have read the paper and cited it in the manuscript. **Please see Lines 105-106.**

2. *Line 288. The authors use "target observation" here, but in the introduction part they used "targeted observation". Please unify the usage.*
**Response:** We have modified the "target observation" to "targeted observation". **Please see Line 305**. We also checked its usages throughout the paper.

3. *Line 290. When the "cost-effective" first appeared in the manuscript, I did not quite understand what it means. More explanations should be added here.*
**Response:** Sorry for confusing the reviewer. The "cost-effective" means assimilating the observations obtained from fewer meteorological stations can lead to higher PM2.5 forecasting skills. This kind of station network can be taken as cost-effective stations because it provides sensitive observations to the PM2.5 forecasts in the economic fashion. The explanations have been added in the revised manuscript. **Please see Lines 308-311.**

4. *Line 323, when determining the sensitive areas, the authors should clarify here that CNOP-type initial errors are superimposed on the ground meteorological fields in the "truth run".*
**Response:** The CNOP-type initial errors which include wind, temperature and water vapor mixing ratio components at the ground level are calculated for each of the 48 PM2.5 forecasts in the "truth run" with the application of WRF and its adjoint model by using the SPG2 solver (see section 2). We have rephrased the sentence in the revised manuscript. **Please see Lines 342-344.**

5. *Line 420, the 110ºE~120Eº should be 110ºE~120º Also the 34N~36N.*
**Response:** We have corrected "110E~120E" to "110ºE~120ºE". We have also corrected "34N~36N" to "34ºN~36ºN". **Please see Line 427.**

6. *Line 705, it is recommended to mention in the section 6 that the improvements are based on the OSSEs, which means the simulated observations from ERA5 are assimilated to the control run to show the effectiveness of the newly refined station observations. However, how the improvements will be when the real observations from the refined station network are assimilated still needs further studies.*
**Response:** We thank the reviewer's suggestion. As we showed on Lines 295-315 in the

manuscript, to identify the sensitive area of the ground meteorological field in each forecast, we adopt the idea of Lorenz (1965) and take the better simulation initialized by ERA5 as "truth run" and the simulation initialized by GFS forecast data as "control run", where these two simulations have the same emission inventory and use the same model; so the difference between them reflect the sensitivities of forecast uncertainties of PM2.5 concentrations on the accuracy of initial meteorological field. When we compute the CNOP-type initial perturbation superimposed on the better simulation initialized by ERA5, it can be regarded as an approximation to the most sensitive initial error and the sensitive area identified by such CNOP-type error can be regarded as an approximation to the real sensitive area. If the approximate sensitive area is valid, assimilating the additional observations in the sensitive area of control forecast will make the updated forecasts approach to the truth run.

Although the present study is associated with hindcasts of PM2.5, it is still difficult to obtain the meteorological observations from the Monitor Center; therefore, we can only assimilate the simulated observations (i.e. the ERA5 data) to the control run to show the effectiveness of the cost-effective observation network. The effectiveness is verified by examining whether a forecast (i.e. the simulation initialized by GFS) after assimilating the observations from the cost-effective station network will be much closer to the good simulation (i.e. the simulation initialized by ERA5). If the cost-effective station network is useful along this thinking, it can be inferred that assimilating real observations from the cost-effective stations to the meteorological initial field in the control forecast would improve the meteorological field forecasting and then the PM2.5 forecasting greatly against the observations. The relevant discussions have been added in the revised manuscript. **Please see Lines 732-741.**

7. *The boundary layer height is also an important meteorological variable for PM2.5 forecasts. Why do not the authors consider the perturbation of this variable in the study?*

**Response:** The CNOP in the present study only considers the sensitivity from initial uncertainties. We agree with the reviewer that the boundary layer height is an important meteorological variable for PM2.5 forecasts. Since the boundary layer simulation is more influenced by the parameterization in the WRF model (Chen et al., 2017; Mohan and Gupta, 2018), to study the role of boundary layer uncertainties in yielding the PM2.5 forecast uncertainties, an extension of the CNOP method, CNOP-parametric perturbation (CNOP-P; Mu et al., 2010) or nonlinear forcing singular vector (NFSV, Duan and Zhou, 2013), can be used to identify the sensitivities of boundary layer uncertainties. The related discussions have been added in the revised manuscript. **Please see Lines 755-764.**

**References:**

Chen, D., Xie, X., Zhou, Y., Lang, J., Xu, T., Yang, N., Zhao, Y., Liu, X., 2017. Performance evaluation of the wrf-chem model with different physical

parameterization schemes during an extremely high PM2.5 pollution episode in Beijing. Aerosol Air Qual. Res. 17 (1), 262–277.

Duan, W., and Zhou, F., 2013. Non-linear forcing singular vector of a two-dimensional quasi-geostrophic model. Tellus, 65(18452), 256-256.

Mohan, M. and Gupta, M., 2018. Sensitivity of PBL parameterizations on PM10 and ozone simulation using chemical transport model WRF-Chem over a sub-tropical urban airshed in India. Atmospheric Environment, 185, 53-63.

Mu, M., Duan, W. S., Wang, Q., and Zhang, R., 2010. An extension of conditional nonlinear optimal perturbation approach and its applications, Nonlin. Processes Geophys., 17(2), 211-220.

**Response to Reviewer #2:**

The manuscript entitled "An approach to refining the ground meteorological observation stations for improving PM2.5 forecasts in Beijing-Tianjin-Hebei region" introduced an approach to refine the ground stations by identifying the sensitive areas for targeted observations. The study is highly related to the studies of predictability, target observation and data assimilation. And it provides a scientific guidance on optimizing the ground stations. I believe the approach is not only useful for air quality forecasts, but can also be used to the forecasts of extreme weather events. Nevertheless, there is a gap between publication and the manuscript in current version. I hope the following comments will help authors improve the manuscript.

**Response:** We thank your appreciations.

**Specific comments:**
*1. Line 42. There are a great many publications addressing the meteorological conditions on PM2.5 variations, but the authors only cite one, which is not enough. More references are needed here.*
**Response:** We thank the reviewer's suggestions. We have added the references (Lou et al., 2019; Chen et al., 2020) in the revised manuscript. **Please see Lines 43.**

*2. Line 68. "assimilating more observations may not necessarily lead to much*

*higher forecast benefits." References are needed here.*

**Response:** We have added the references here (Li et al., 2010; Liu et al., 2021). **Please see Line 69.**

*3. Line 75. How are the worse forecast skills possible when the sensitivities are low? Please provide a detailed explanation here.*

**Response:** We have added a detailed explanation in the revised manuscript. **Please see Lines 77-78.** Theoretically, if the observations in the area where the forecast is not sensitive to the initial errors are assimilated, the forecast skills might be slightly improved or neutral. However, in realistic forecasts, the imperfect assimilation procedure or the unresolved scales and processes in the model may induce additional errors and lead to the worse forecasts when the observations in the area where the forecast is not sensitive to the initial errors are assimilated (Janjic et al., 2018). For example, in Yu et al. (2012), removing the initial error in the area that is not the most sensitive area could worsen the prediction results of ENSO. That emphasized the importance of identifying the most sensitive area and suggests that additional observations should be assimilated more carefully in this sense.

*4. Line 195-202. The descriptions are insufficient and confuse me. Please add more details and make it clear.*

**Response:** Sorry for confusing the reviewer. We have rewritten the paragraph and make it clearer. **Please see Lines 205-219.**

"The spectral projected gradient 2 (SPG2) method is used to solve the optimization problem in Eq. (3). It is noted that the SPG2 algorithm is generally designed to solve the minimum value of nonlinear function (cost function) with an initial constraint condition, and the gradient of cost function with respect to the initial perturbation represents the descending direction of searching for the minimum of the cost function. Therefore, in this study, we have to rewrite the cost function Eq.(3) as $J'(\delta x_0^*) = \min\limits_{\delta x_0^T C_1 \delta x_0 \leq \beta} -[M(x_0 + \delta x_0) - M(x_0)]^T C_2 [M(x_0 + \delta x_0) - M(x_0)]$ and the WRF adjoint model is used to compute the gradient of the cost function. Specially, to calculate the CNOP, a first guess initial perturbation, $\delta x_0^{(0)}$, is projected into the constraint condition and superimposed on the initial state $x_0$ of the WRF model. After a forward integration of the WRF, the value of the cost function, i.e. $-[M(x_0 + \delta x_0^{(0)}) - M(x_0)]$, can be obtained. Then, with the adjoint model of the WRF, the gradient of the cost function with respect to the initial perturbation, $g(\delta x_0^{(0)})$, is calculated. Theoretically, the gradient presents the fastest descending direction of the cost function. However, in realistic numerical experiments, the gradient presents the fast-descending direction but not necessarily the fastest, so we need more iterations. After iteratively forward and backward integrations of the WRF model governed by the SPG2 algorithm, the initial perturbation is optimized and updated until the convergence condition is satisfied,

where the convergence condition is $\left\| P\left(\delta x_0^{(p)} - g\left(\delta x_0^{(p)}\right)\right) - \delta x_0^{(p)} \right\|_2 \leq \varepsilon_1$ and $\varepsilon_1$ is an extremely small positive number, $P(\delta x_0^{(p)})$ projects the initial perturbation to the constraint condition. Finally, the CNOP $\delta x_0^{(p)}$ can be obtained. The flow chart of the CNOP calculation is shown in Figure 2. For further details of the SPG2 algorithm, the readers can be referred to Birgin et al. (2001)."

*5. Line 313. Please clarify that the real "meteorological" observations are not in public archive, because in section 3.1, the authors have compared the simulations with the observed PM2.5 concentrations.*

**Response:** We have clarified that the real meteorological observations are not available in public archive in the revised manuscript. **Please see Line 333.** The sentence have been corrected into "Since the real meteorological observations are not in public archive, the "additional observations" are correspondingly taken from the initial field of the truth run (i.e. the ERA5 data) and called as "simulated observations" according to the OSSEs.".

*6. Line 322. Is the CNOP-type initial error that what has been described in section 2.3? It is suggested to add a detailed description on what variables the CNOP-type errors have contained here.*

**Response:** Yes, the CNOP-type initial error is what has been described in section 2.3. We have added a detailed description of CNOP-type error here. The sentence has been corrected into "the CNOP-type initial errors which includes wind, temperature and water vapor mixing ratio components at the ground level are calculated for each of the 48 PM2.5 forecasts with the application of WRF and its adjoint model by using the SPG2 solver (see section 2). **Please see Lines 342-344.**

*7. Line 349. "the area with larger values of TME can be regarded as the sensitive areas". It is ambiguous. Is there a threshold for the definition of sensitive area or just determined subjectively?*

**Response:** The TME is applied to measure the comprehensive sensitivity of PM2.5 forecast uncertainties on initial meteorological perturbations. When we identify the essential observational network, we take the 3% as the threshold to determine the sensitive area. Then a total of 424 sensitive grid points is obtained. We select the 3% as the threshold here because the number 424 of sensitive grid points is close to the number of 481 of the meteorological stations within and surrounding the BTH region. **Please see Lines 411-414.**

*8. Line 453. "the essential stations can indeed provide additional observations that help increase the skill of the PM2.5 forecasts, in comparison to other constructed stations but not in the sensitive grids". The authors did not do any comparison experiments to show the improvements are higher than assimilating the station*

*observations which are not in the sensitive grids. How can they get such conclusions?*
**Response:** We have corrected the sentence into "It is clear that the essential stations can indeed provide additional observations that help increase the skill of the PM2.5 forecast in the BTH much significantly". **Please see Line 473.**

*9. As shown in Figure 7 (a1, a2), assimilating the observations will lead to worse forecasts since the AEv and AEM are negative. It is hard to understand. Why will assimilating the observations will lead to worse forecasts?*
**Response:** The negative PM2.5 forecast skills occurred at the AF initialized at 20:00 on Nov 18th 2016. For the AF initialized at 20:00 at Nov 18th 2016, the PM2.5 concentration in the truth run increases from 139.5 $\mu g\, m^{-3}$ to 151.5 $\mu g\, m^{-3}$ averaged over the BTH region; while the control run forecasts the PM2.5 concentration of 159.6 $\mu g\, m^{-3}$ averaged over the BTH region, 20.1 $\mu g\, m^{-3}$ higher than the PM2.5 concentration in the truth run. When all the constructed station observations are assimilated, the PM2.5 concentration averaged over the BTH region is 153.11 $\mu g\, m^{-3}$ at the forecast time, much closer to the truth run. However, the improvements in the BTH region are uneven (Figure 2), and the number of grids showing negative improvements overweigh those showing positive improvements, which results in a negative AEv and AE$_M$.

[Figure]

Figure 2 The improvements of PM2.5 forecast skills when all the constructed station observations are assimilated.

*10. Line 703. "It is clear that assimilating the fewer observations can lead to higher PM2.5 forecast skills". It is inaccurate. It is suggested to rephrase it more carefully.*
**Response:** We have corrected the sentence into "It is clear that assimilating the fewer sensitive observations can lead to higher PM2.5 forecast skill". **Please see Line 723.**

*11. It is suggested to mention the limitation of the study in section6 that the results are based on OSSEs. If the real observations are available, how the refined*

***station observations help improve the air quality forecasts deserve deeper studies.***

**Response:** We thank the reviewer's suggestion. As we showed on Lines 295-315 in the manuscript, to identify the sensitive area of the ground meteorological field in each forecast, we adopt the idea of Lorenz (1965) and take the better simulation initialized by ERA5 as "truth run" and the simulation initialized by GFS forecast data as "control run", where these two simulations have the same emission inventory and use the same model; so the difference between them reflect the sensitivities of forecast uncertainties of PM2.5 concentrations on the accuracy of initial meteorological field. When we compute the CNOP-type initial perturbation superimposed on the better simulation initialized by ERA5, it can be regarded as an approximation to the most sensitive initial error and the sensitive area identified by such CNOP-type error can be regarded as an approximation to the real sensitive area. If the approximate sensitive area is valid, assimilating the additional observations in the sensitive area of control forecast will make the updated forecasts approach to the truth run.

Although the present study is associated with hindcasts of PM2.5, it is still difficult to obtain the meteorological observations from the Monitor Center; therefore, we can only assimilate the simulated observations (i.e. the ERA5 data) to the control run to show the effectiveness of the cost-effective observation network. In our study, the effectiveness is verified by examining whether a forecast (i.e. the simulation initialized by GFS) after assimilating the observations from the cost-effective station network will be much closer to the good simulation (i.e. the simulation initialized by ERA5). If the cost-effective station network is useful along this thinking, it can be inferred that assimilating real observations from the cost-effective stations to the meteorological initial field in the control forecast would improve the meteorological field forecasting and then the PM2.5 forecasting greatly against the observations. The relevant discussions have been added in the revised manuscript. **Please see Lines 732-741.**

**References:**

Janjić, T, Bormann, N, Bocquet, M, Carton, JA, Cohn, SE, Dance, SL, Losa, SN, Nichols, NK, Potthast, R, Waller, JA, Weston, P., 2018. On the representation error in data assimilation, Q J R Meteorol Soc. 144: 1257– 1278.

Li, X., Zhu, J., Xiao, Y., and Wang, R., 2010. A Model-Based Observation-Thinning Scheme for the Assimilation of High-Resolution SST in the Shelf and Coastal Seas around China, Journal of Atmospheric and Oceanic Technology, 27(6), 1044-1058.

Liu C., Zhang, S., Gao, Y., Wang, Y. and coauthors, 2021. Optimal estimation of initial concentrations and emission sources with 4D-Var for air pollution prediction in a 2D transport model. Science of the Total Environment, 773, 145580.

Lou, M., Guo, J., Wang, L., Xu, H., Chen, D., Miao, Y., Lv, Y., Li, Y., Guo, X., Ma, S., Li, J., 2019. On the relationship between aerosol and boundary layer height in summer in China under different thermodynamic conditions. Earth Space Sci. 6 (5), 887–901.

Yu, Y., Mu, M., Duan, W., Gong, T., 2012. Contribution of the location and spatial pattern of initial error to uncertainties in El Niño predictions. Journal of Geophysical Research, 117, C06018.

**Response to Community comments:**

**General comments:**

*The authors present an approach to refine the ground meteorological stations in order to improve the regional air quality forecasts. Based on the sensitive areas for targeted observation, the authors first identified the essential stations, then scattered the other stations with different distances, finally a cost-effective observation network is provided. The refinement of the ground stations is a desirable one and the method is described in a clear and logical manner. Apart from the comments posted on the website by two referees, I only have a few minor comments.*
**Response:** We thank your appreciations.

**Specific comments:**

*1. What are the compositions of PM2.5 matter considered in the model?*
**Response:** The components of PM2.5 simulation here include black carbon, organic carbon, secondary inorganic aerosol (sulfate, nitrate, ammonium) and primary PM2.5 emitted directly from various sources. The compositions of PM2.5 matter considered in the model have been added in the revised manuscript. **Please see Lines 142-144.**

*2. Is it the domain setup in WRF the same with NAQPMS? Please clarify.*
**Response:** The WRF model is configured with the same horizontal and vertical grid structure with the NAQPMS model. The details have been added in the revised manuscript. **Please see Lines 150-151.**

*3. The boundary layer height is a key meteorological variable that affects the regional PM2.5 concentration, but the authors do not consider it in the definition of CNOP-type errors. The reasons why the BLH is not considered should be mentioned.*
**Response:** We agree with the reviewer that the boundary layer height is a key meteorological variable that affects the regional PM2.5 concentration forecasts. The CNOP in the present study only considers the sensitivity from initial uncertainties. Since the boundary layer simulation is more influenced by the parameterization in the WRF model (Chen et al., 2017; Mohan and Gupta, 2018), to study the role of boundary layer uncertainties in yielding the PM2.5 forecast uncertainties, an extension of the

CNOP method, CNOP-parametric perturbation (CNOP-P; Mu et al., 2010) or nonlinear forcing singular vector (NFSV, Duan and Zhou, 2013), can be used to identify the sensitivities of boundary layer uncertainties. Future studies addressing the boundary layer uncertainties using the extensions of CNOP method and the design on its relevant observation network can be expected. The related discussions have been added in the revised manuscript. **Please see Lines 755-764.**

4. *I am curious about how much improvements will be when the cost-effective stations are removed from the all the constructed stations. It is suggested to add some experiments in the manuscript. At least the authors could take some forecasts as examples to show the differences.*

**Response:** We thank the reviewer's suggestions. The CNOP-type error represents the initial error that results in the largest forecast error in the verification area at the verification time. The CNOP-type error considers the interaction among the errors on spatial grid points and in this situation, the errors on the grid points with large amplitude of the CNOP-type error contribute much more to the final prediction error. When we sort the spatial grid points with a decreasing order according to the amplitude of the error and choose the first 3% grid points as the essential grid points, the interactions between these grid points are remained, so that it is assumed that assimilating the observations on these grid points may contribute more to the improvements of forecast skills. Based on a series of OSSEs, it is verified that assimilating the essential or cost-effective observations can indeed improve greatly the PM2.5 forecasts. Specifically, when the 279 cost-effective station observations are assimilated for the AF in section 5, they achieve an overall 41.11% the improvement of PM2.5 forecasting skills, which explains 99% the improvement when assimilating constructed station observations; furthermore, when the cost-effective station observations are removed from all the constructed station observations, the number of the rest station observations is 77 smaller than that of the cost-effective station observations and the assimilation of these observations explains much less, which is 70% the improvement obtained by assimilating all constructed station observations. To be specially emphasized, for the DF presented in section 5, when the simulated observations from the 241 cost-effective station observations are assimilated, it results in an improvement of 47.55% of PM2.5 forecasting skills, even 1.7% higher than the improvement of assimilating all constructed station observations; however, when the cost-effective station observations are removed, assimilating the rest 240 station observations would only result in an improvement of 22.60% PM2.5 forecasting skill. Obviously, although the number of rest station observations is almost the same with the cost-effective station observations, the improvement of PM2.5 forecasting skills is less than half of the improvements obtained by assimilating the cost-effective station observations.

Totally, assimilating the cost-effective station observation will lead to much higher PM2.5 forecasting skills than assimilating the rest observations, which emphasizes the important role of the cost-effective station observations in improving the PM2.5 forecast skills.

In the current version of manuscript, we have considered the interactions among

the errors on spatial grid points so that we choose the grid points with large amplitude of the CNOP-type errors, which will have large impact on final prediction result, to construct the cost-effective observation network. So the cost-effective observation network is constructed under the consideration of the interactions among the errors on spatial grid points. Moreover, it is verified numerically that assimilating these cost-effective station observations can achieve the PM2.5 forecasting skills comparable to that obtained by assimilating all ground station observations. We think it is enough to demonstrate the effectiveness of cost-effective station observations. However, if we assimilate the rest station observations, since the interactions among errors on spatial grid points are considered, the impact of the cost-effective station observations could also be reflected in the improvement of forecasting skill obtained by the assimilation of rest station observations. That's why assimilating the rest station observations could explain roughly 70% the improvement obtained by assimilating all constructed station observations in the AF shown in section 5. In this situation, it is hard to distinguish the role of cost-effective station observation. Therefore, even though we have implemented the experiments suggested by the reviewer, to avoid confusing the readers, we think it is better to keep the original structure and do not include the additional experimental results in the revised manuscript.

5. *A series of OSSEs is designed to verify the effectiveness of refined stations, due to the unavailable of real meteorological observations. It is suggested to add more discussions on how future work could use real observations.*

**Response:** We thank the reviewer's suggestion. As we showed on Lines 295-315 in the manuscript, to identify the sensitive area of the ground meteorological field in each forecast, we adopt the idea of Lorenz (1965) and take the better simulation initialized by ERA5 as "truth run" and the simulation initialized by GFS forecast data as "control run", where these two simulations have the same emission inventory and use the same model; so the difference between them reflect the sensitivities of forecast uncertainties of PM2.5 concentrations on the accuracy of initial meteorological field. When we compute the CNOP-type initial perturbation superimposed on the better simulation initialized by ERA5, it can be regarded as an approximation to the most sensitive initial error and the sensitive area identified by such CNOP-type error can be regarded as an approximation to the real sensitive area. If the approximate sensitive area is valid, assimilating the additional observations in the sensitive area of control forecast will make the updated forecasts approach to the truth run.

Although the present study is associated with hindcasts of PM2.5, it is still difficult to obtain the meteorological observations from the Monitor Center; therefore, we can only assimilate the simulated observations (i.e. the ERA5 data) to the control run to show the effectiveness of the cost-effective observation network. In our study, the effectiveness is verified by examining whether a forecast (i.e. the simulation initialized by GFS) after assimilating the observations from the cost-effective station network will be much closer to the good simulation (i.e. the simulation initialized by ERA5). If the cost-effective station network is useful along this thinking, it can be inferred that assimilating real observations from the cost-effective stations to the meteorological

initial field in the control forecast would improve the meteorological field forecasting and then the PM2.5 forecasting greatly against the observations. The relevant discussions have been added in the revised manuscript. **Please see Lines 732-741.**

**References:**
Chen, D., Xie, X., Zhou, Y., Lang, J., Xu, T., Yang, N., Zhao, Y., Liu, X., 2017. Performance evaluation of the wrf-chem model with different physical parameterization schemes during an extremely high PM2.5 pollution episode in Beijing. Aerosol Air Qual. Res. 17 (1), 262–277.

Duan, W., and Zhou, F., 2013. Non-linear forcing singular vector of a two-dimensional quasi-geostrophic model. Tellus, 65(18452), 256-256.

Mohan, M. and Gupta, M., 2018. Sensitivity of PBL parameterizations on PM10 and ozone simulation using chemical transport model WRF-Chem over a sub-tropical urban airshed in India. Atmospheric Environment, 185, 53-63.

Mu, M., Duan, W. S., Wang, Q., and Zhang, R., 2010. An extension of conditional nonlinear optimal perturbation approach and its applications, Nonlin. Processes Geophys., 17(2), 211-220